# Stratospheric Aerosol Extinction Profiles from SCIAMACHY Solar Occultation

Stefan Noël[1], Klaus Bramstedt[1], Alexei Rozanov[1], Elizaveta Malinina[1,2], Heinrich Bovensmann[1], and John P. Burrows[1]

[1]Institute of Environmental Physics, University of Bremen, FB 1, P.O. Box 330440, 28334 Bremen, Germany
[2]now at the Canadian Centre for Climate Modelling and Analysis (CCCma), Environment and Climate Change Canada,Victoria, Canada

**Correspondence:** S. Noël (stefan.noel@iup.physik.uni-bremen.de)

**Abstract.**

The SCIAMACHY (Scanning Imaging Absorption Spectrometer for Atmospheric CHartographY) instrument on ENVISAT provided between August 2002 and April 2012 measurements of solar and Earthshine spectra from the UV to the SWIR spectral region in multiple viewing geometries.

We present a new approach to derive stratospheric aerosol extinction profiles from SCIAMACHY solar occultation measurements based on an onion peeling method similar to the Onion Peeling DOAS (Differential Optical Absorption Spectroscopy) retrieval, which has already been successfully used for the derivation of greenhouse gas profiles. Since the retrieval of aerosol extinction requires as input measured transmissions in absolute units, an improved radiometric calibration of the SCIAMACHY solar occultation measurements has been developed, which considers various instrumental and atmospheric effects specific to solar occultation.

The aerosol extinction retrieval can in principle be applied to all wavelengths measured by SCIAMACHY. As a first application, we show results for 452 nm, 525 nm and 750 nm. The SCIAMACHY solar occultation time series has been processed, covering a latitudinal range of about 50–70°N. Reasonable aerosol extinctions are derived between about 15 and 30 km with typically larger uncertainties at higher altitudes due to decreasing aerosol extinction.

Comparisons with collocated SAGE-II and SCIAMACHY limb aerosol data products revealed a good agreement with essentially no mean bias. However, dependent on altitude differences of up to $\pm20$–30% to SAGE-II at 452 nm and 525 nm are observed. Similar results are obtained from comparisons with SAGE-III.

SCIAMACHY solar occultation data at 750 nm have been compared with corresponding SAGE-III, OSIRIS and SCIAMACHY limb results. The agreement with SCIAMACHY limb data at 750 nm is within 5–20% between 17 and 27 km. SAGE-III and OSIRIS show at this wavelength and altitude range on average about 40% and 25% smaller values, with some additional 10–20% modulation with altitude.

The altitude variations in the differences are mainly caused by systematic vertical oscillations in the SCIAMACHY occultation data of up to 30% below about 25 km. These oscillations decrease to amplitudes below 10% with increasing number of collocations and are no longer visible in monthly anomalies.

Major volcanic eruptions as well as occurrences of PSCs can be identified in the time series of aerosol extinction data and related anomalies. Influences of the Quasi-Biennial-Oscillation (QBO) are visible above $25\,\mathrm{km}$.

# 1  Introduction

Stratospheric aerosols play an important role in climate as they affect radiative forcing either by scattering and absorption of light (direct effect) or by their impact on clouds and ozone (indirect effect). Especially, aerosols affect the creation of polar

stratospheric clouds (PSCs) on which surfaces $O_3$ depletion takes place.

The main constituents of stratospheric aerosols are sulfuric acid ($H_2SO_4$) and (liquid) water ($H_2O$). Sulfuric acid is mostly produced from oxidation of carbonyl sulfide (OCS) and sulfur dioxide ($SO_2$). OCS has mostly marine origin while $SO_2$ mainly originates from volcanic eruptions, biomass burning (both natural and anthropogenic origin) and fossil fuel combustion. The main transport path of OCS and $SO_2$ in the stratosphere during volcanic quiescent periods is the tropical upwelling. In addition,

anthropogenic $SO_2$ from fossil fuel combustion is transported to the stratosphere via the Asian monsoon (Randel et al., 2010) while pyrocumulus events represent a transport mechanism for biomass burning products. Large amounts of $SO_2$ can be directly injected into the stratosphere by strong volcanic eruptions.

Information about stratospheric aerosols can be derived e.g. from ground based lidars or in-situ balloon and aircraft measurements. However, these usually have a limited spatial and temporal coverage. Global measurements of stratospheric aerosols

are only possible with satellite based instruments, see Table 1.

These started in the 1970ies with the suite of SAM (Stratospheric Aerosol Measurement) and SAGE (Stratospheric Aerosol and Gas Experiment) instruments (see e.g. Chu and McCormick, 1979; Kent and McCormick, 1984; McCormick, 1987; Osborn et al., 1989; Russell and McCormick, 1989; Thomason and Taha, 2003; Thomason et al., 2008).

From 1991 until 2001 HALOE (Halogen Occultation Experiment on the Upper Atmosphere Research Satellite (UARS);

see e.g. Russell et al., 1993; Lee et al., 2001) performed occultation measurements from which among other atmospheric constituents also aerosol extinctions were derived. Aerosol extinction profiles were also provided by the Polar Ozone and Aerosol Measurement (POAM) II and III between 1993 and 2005 (see e.g. Bevilacqua et al., 1995; Randall et al., 1996, 2001). These were accompanied between 2002 and 2012 by the ENVISAT instruments GOMOS (Global Ozone Monitoring by Occultation of Stars; Kyrölä et al., 2010), MIPAS (Michelson Interferometer for Passive Atmospheric Sounding; Fischer et al.,

2008) and SCIAMACHY (Scanning Imaging Absorption Spectrometer for Atmospheric CHartographY; see e.g. Gottwald and Bovensmann, 2011). Currently, OSIRIS (Optical Spectrograph and InfraRed Imager System, see e.g. Llewellyn et al., 2004; Bourassa et al., 2012; Rieger et al., 2014, 2019), ACE-MAESTRO (the Atmospheric Chemistry Experiment Measurement of Aerosol Extinction in the Stratosphere and Troposphere Retrieved by Occultation instrument; McElroy et al., 2007; Sioris et al., 2010), CALIOP (Cloud-Aerosol Lidar with Orthogonal Polarization Lidar; Winker et al., 2007; Vernier et al., 2011),

OMPS (Ozone Mapping Profiler Suite; Jaross et al., 2014; Loughman et al., 2018; Chen et al., 2018), and SAGE-III (from ISS; Cisewski et al., 2014) are still operational and deliver data on stratospheric aerosols. Note that the CALIOP instrument is an

active sounder, which is primarily intended for tropospheric aerosol extinction measurements. However, recently also results from stratospheric aerosol extinction retrievals are available, see (Thomason et al., 2007; Pitts et al., 2018; Kar et al., 2019).

These satellite instruments measure via different viewing geometries, all having advantages and drawbacks. Occultation instruments can more or less directly measure aerosol extinction (i.e. the sum of scattering and absorption of light). Limb data are usually more difficult to be analysed, see e.g. Malinina et al. (2018), who derived particle size distributions from SCIAMACHY limb measurements.

In the following we describe a new method to derive stratospheric aerosol extinction profiles from SCIAMACHY solar occultation data. This method is in principle able to derive aerosol extinction profiles at any wavelength measured by SCIA-MACHY. To demonstrate this, we concentrate in the current study on selected wavelengths in the visible / near infrared spectral region where also suitable correlative data sets (SAGE-II and SCIAMACHY limb) are available, namely 452 nm, 525 nm and 750 nm.

The manuscript is organised as follows: Section 2 lists the input data used in this study. In Section 3 the SCIAMACHY occultation data are described with the focus on the newly developed radiometric calibration, which is the largest challenge in this context. Section 4 explains the aerosol extinction retrieval method. The corresponding retrieval results and their first validation are then shown in Section 5. In Section 6 we show time series of the aerosol extinction data. The conclusions are summarised in Section 7. Some details on the methods used in this study are given in the appendix.

## 2 Used data

### 2.1 SCIAMACHY spectra

The SCIAMACHY solar occultation data used in this study were extracted from the SCIAMACHY Level 1 Version 8 product with all calibrations applied except for polarisation correction as solar irradiances are unpolarised. Additional pointing corrections as described in Bramstedt et al. (2017) have been applied such that the tangent height knowledge is better than 26 m. These radiance measurements are then converted into transmissions using additional corrections as will be described in detail in section 3.

### 2.2 ECMWF ERA Interim

ECMWF ERA Interim model data (Dee et al., 2011) are used in the retrieval to account for pressure and temperature profiles at the time and location of the measurements (see section 4.1). These data are available every 6 hours on a 0.75° horizontal grid and on 60 altitude levels.

### 2.3 SAGE-II profiles

The SAGE-II instrument performed solar occultation measurements from 1984 to 2005 and provided aerosol extinction profiles at several wavelengths (386 nm, 452 nm, 525 nm, 1020 nm) as well as profiles of $O_3$, $NO_2$ and $H_2O$. In this study we use

SAGE-II V7.00A sunset aerosol extinction data (Damadeo et al., 2013) from the overlap period with SCIAMACHY (2002 to 2005) at 452 nm and 525 nm for comparisons. We selected collocated data within a maximum spatial distance of 800 km. Since both data sets are based on sunset measurements, the actual time differences are always smaller than 1 h. These criteria result in 700 collocations.

## 2.4 SAGE-III profiles

The SAGE-III instrument flown on Meteor-3M provided between 2002 and 2005 aerosol extinction profiles from solar occultation measurements at nine wavelengths from 384.3 nm to 1545.2 nm. Here, we use SAGE-III V4.0 sunset aerosol extinction data (Thomason and Taha, 2003) from the overlap period with SCIAMACHY (2002 to 2005) at 449 nm, 520 nm and 755 nm. As for SAGE-III, we took collocated data within 800 km distance. The maximum time offset is below 1 h. The number of achieved collocations is 5505.

## 2.5 OSIRIS limb aerosol

The OSIRIS instrument on ODIN provides limb aerosol extinctions at 750 nm since 2001, see Rieger et al. (2014). For the comparisons with SCIAMACHY solar occultation data we use product V7.0 (Rieger et al., 2019) and collocations with a maximum spatial/time difference of 800 km/9 h. There are 12108 collocations, but essentially no collocations between November and February. Relaxation of the collocation criteria, e.g. to 1000 km/12 h, does not help here. This is because OSIRIS observation geometry and orbit parameters result in measurement gaps at winter high latitudes.

## 2.6 SCIAMACHY limb aerosol

The SCIAMACHY limb aerosol extinction product V1.4 was obtained by using the algorithm described in Rieger et al. (2018). It comprises stratospheric profiles derived from SCIAMACHY limb measurements at 750 nm. The data have been filtered according to the recommendations given by the data providers in the accompanying README file; especially, invalid data and data points with a vertical resolution larger than 7 km or aerosol extinctions exceeding $0.1\,\mathrm{km^{-1}}$ have not been used. The spatial collocation criterion is the same as for SAGE-II, but we used a maximum time difference of 10 h. This is necessary to achieve also collocations in summer.

# 3 SCIAMACHY occultation

## 3.1 Measurements

The SCIAMACHY instrument performed measurements in nadir, limb and solar/lunar occultation geometry covering continuously the spectral range from about 214 nm to 1750 nm and two additional bands at around 2000 nm and 2400 nm.

SCIAMACHY performed solar occultation measurements every orbit in the Northern hemisphere (between about 50°N to
70°N) during time of (local) sunset. During such a measurement SCIAMACHY observed the (apparently) rising sun through
the atmosphere with the following typical measurement sequence (see also Fig. 1, from Noël et al., 2016):

1. Perform a sequence of up- and downward scans around an altitude of 17.2 km until the centre of the (un-refracted) sun
   reaches this altitude.

2. Switch-on the so-called sun follower in azimuthal direction to horizontally align the viewing direction to the intensity
   centre of the sun.

3. Follow the rising sun while scanning vertically around the (predicted) centre of the sun until about a tangent altitude of
   100 km.

Above 100 km either special solar calibration measurements are performed or the scan over the sun is continued up to about
250 km. In this study, we concentrate on data below 100 km, such that all available solar occultation measurements can be
used.

## 3.2 Transmissions

The aerosol extinction retrieval (see below) requires as input atmospheric transmissions. In order to derive these transmissions,
the individual SCIAMACHY spectra are in a first step normalised to a reference spectrum obtained at a high tangent altitude
of about 90 km.

This is done independently for up- and downward scans. With this, all possibly erroneous multiplicative calibration factors
(e.g. most degradation effects or systematic errors in radiometric calibration) cancel out, which is why occultation measure-
ments are sometimes called 'self-calibrating'.

However, this is not really the case for SCIAMACHY because of the scan over the sun. The width of the instantaneous field
of view (IFOV) of SCIAMACHY is in solar occultation mode about 0.7°, the height about 0.045°. As the diameter of the sun is
about 0.5°, this implies a strongly varying signal over the scan as different parts of sun are seen at each readout. Furthermore,
refraction effects and additional problems due to e.g. mispointing and jumps in the signal when switching on the sun follower
need to be taken into account. This is explained in the following subsections.

### 3.2.1 Radiometric calibration / scan correction

The largest impact on the measured signal is related to the area of the sun seen during each readout, which varies over the scan.
This is mainly a geometric effect, which is illustrated in Fig. 2. Depending on the vertical position of the SCIAMACHY IFOV
relative to the centre of the sun different areas of the IFOV are illuminated. The measured signal then varies approximately
with the size of the illuminated area.

The right plots of Fig. 3 show this varying signal for the reference scan at high tangent altitudes, where atmospheric absorp-
tion and refraction are small and can be neglected. All data are normalised to the (interpolated) maximum signal of the scan. In

the top right figure the signal is shown for an upward and a downward scan as function of geometric tangent altitude. Because the sun is (relative to the instrument) rising during the measurement, an upward scan covers a larger altitude range than a downward scan. However, as can be seen in the bottom right figure, the variation of the signal becomes very similar when plotted as function of angular (vertical) distance from the centre of the sun. The thick black line in this figure shows the result of a simple geometrical model of the varying area when assuming a circular sun disk of diameter $0.26°$ with homogeneous brightness. The overall shape of the measurements is reproduced quite well by the black line; the deviations are caused by the facts that 1) the real sun does not have the same brightness everywhere (mainly because of limb darkening effects) and 2) the measured signal is an integral over the IFOV in vertical direction ($0.045°$ correspond to about $2.6\,\mathrm{km}$) which smears out the black curve along the x axis.

The left plots of Fig. 3 show the corresponding measured transmissions for various scans at lower tangent altitudes as function of tangent height (top left) and distance to the sun centre (bottom left). The normalisation is the same as for the right plots, i.e. all upward/downward scans (even/odd numbers) are normalised to the maximum value of of the reference measurement (green/red curve in the right plot). Due to increased atmospheric absorption and scattering the transmissions decrease at lower altitudes. In addition, as can be seen in the bottom left plot, the maximum signal of the scan shifts to the right with decreasing altitude due to increasing refraction.

The plots of Fig. 3 also show that the measured signal for one scan is not symmetrical relative to the sun centre, i.e. the signal drops to zero only on one side. This is because the position and elevation rate of the sun assumed in the commanding of the measurement was derived from predicted orbital information. This results in a scan which is not exactly centred on the (true) sun. This may also lead to azimuthal offsets (see right plot of Fig. 2), which are corrected by use of the sun follower (see above), but this can introduce jumps in the signal at altitudes around $17\,\mathrm{km}$ which require special treatment (see Appendix A).

To correct for the scan effect, we define a (numerical) sun shape function $S$, which is the interpolated measured transmission ($T^\mathrm{m}$) for a scan around a reference altitude of about $90\,\mathrm{km}$ as function of angular distance from the centre of the sun ($\alpha$), as shown in Fig. 3 (bottom right). This is done for each measurement and independently for both up- and downscan in order to reduce possible systematic effects caused by the scan direction. Note that $S$ describes the shape of the sun without atmospheric effects like refraction or influences of aerosol or clouds which can be neglected at $90\,\mathrm{km}$.

To account for refraction effects we use a simple model similar to the one used in the SAGE-II project (Damadeo et al., 2013), see Fig. 4. It is assumed that refraction occurs only at the tangent point with the basic parameter being the bending angle ($\delta$). This bending angle decreases with altitude and is essentially a function of pressure. In the stratosphere, the overall altitude variation of $\delta$ can therefore be described by an exponential function of tangent height $z$:

$$\delta = \exp(a + b\,z) \tag{1}$$

The parameters $a$ and $b$ depend on atmospheric conditions (and also on wavelength) and are different for each measured profile. $b$ is typically negative, as refraction effects decrease with altitude. Therefore we determine these parameters from the measurements (see Appendix B). From these we then get for each measurement the bending angle $\delta$ from which we calculate

the distance $\alpha$ of the observed point on the sun to the sun centre via:

$$\alpha = \gamma - \beta - \delta \qquad (2)$$

Here, $\gamma$ is the line-of-sight (LOS) zenith angle, $\beta$ is the direction of the 'true' sun (i.e. without refraction). The latter is essentially the solar zenith angle (SZA) at the satellite position, i.e. $\beta = 180° - \text{SZA}$. The LOS zenith angle $\gamma$ and the solar zenith angle are given in the SCIAMACHY Level 1 product for the centre of the IFOV. As we assume a horizontally homogeneous atmosphere (within the range of one measured profile) azimuthal differences are not relevant in this context. However, as mentioned before, possible azimuthal jumps at lower altitudes need to be considered, see Appendix A.

The expected transmission corresponding to the distance $\alpha$ is then given by the sun shape function $S(\alpha)$ derived from the reference scans (see above). The scan-corrected transmission $T$ as function of tangent altitude $z_i$ for readout $i$ of an occultation measurement is then derived from:

$$T_i = T(z_i) = T_i^{\mathrm{m}}/S(\alpha_i) \qquad (3)$$

### 3.3 Selection of subset of readouts

Prior to the retrieval (see Section 4.1) the measured transmissions need to be interpolated to a fixed altitude grid. Therefore it is sufficient to use only a subset of the measured spectra for this. This subset is basically selected by using readouts with the highest (uncorrected) transmission signal, which corresponds e.g. to the envelope of the data points shown in the top left plot of Fig. 3. As an additional criterion, we only take data points with an altitude difference of $0.5\,\mathrm{km}$ or larger (when starting at the top and then going downwards in altitude). An example showing the results of this procedure is given in section 5.1.

## 4 Retrieval method

The basic idea for the aerosol extinction retrieval is to use a two step approach:

1. Apply the Onion Peeling DOAS (Differential Optical Absorption Spectroscopy) retrieval method to correct the measured transmissions for Rayleigh scattering and gas absorptions.

2. Use an onion peeling method to determine aerosol extinctions from corrected transmissions for different altitude layers, starting with the highest layer.

These two steps are described in more detail in the following sub-sections.

With this approach it is possible to determine aerosol extinctions even at wavelengths where gases absorb (since this absorption is fitted). In addition, the method also delivers profiles of the absorbing gases. However, these derived stratospheric gas profiles (in the present case for $O_3$ and $NO_2$) are not the primary focus of the current study as retrieval settings are optimised for aerosol extinction.

### 4.1 Onion Peeling DOAS Approach

The Onion Peeling DOAS (ONPD) retrieval method has been originally developed to derive stratospheric profiles of greenhouse gases. So far, it has been applied to the retrieval of water vapour, $CO_2$ and methane (Noël et al., 2010, 2011, 2016, 2018). The retrieval method is described in detail in these publications; we therefore give here only a basic summary and the specific settings used in the context of this study.

#### 4.1.1 Description of method

In the ONPD approach the atmosphere is divided into layers. All measured transmission spectra are interpolated to this grid. For each tangent height $j$ a weighting function DOAS fit (see e.g. Coldewey-Egbers et al., 2005) is performed using the following formula:

$$\ln T_j^{\mathrm{interp}} = \ln T_{j,\mathrm{ref}} + \sum_k \sum_i w_{ij,k}\, a_{i,k} + P_j \tag{4}$$

Here, $T_j^{\mathrm{interp}}$ is the (interpolated) measured transmission for tangent height $j$. $T_{j,\mathrm{ref}}$ is a reference transmission derived for the same viewing geometry from a radiative transfer model, in our case SCIATRAN V3.7 (Rozanov et al., 2013) in occultation mode. The index $i$ refers to the atmospheric layers, $k$ to the different absorbers considered in the fit. $w_{ij,k}$ is the relative weighting function, which is also derived by the radiative transfer model. It describes how the (logarithmic) transmission for tangent height $j$ changes if the amount of absorber $k$ is changed by 100% in layer $i$. $a_{i,k}$ is a scalar factor, which describes the actual change of absorber $k$ in layer $i$ relative to the assumptions in the radiative transfer model. Spectrally broadband absorption and scattering (especially due to aerosols) is described by a polynomial $P_j$.

The factors $a_{i,k}$ and the polynomial $P_j$ are fitted for each layer $j$, starting at the top layer and then propagating downwards. In each step the results of the upper layers are taken into account. From the combination of the $a_{i,k}$ scaling factors with the a-priori profiles assumed in the radiative transfer calculations vertical profiles of the absorbers $k$ are derived. These profiles are then vertically smoothed using a boxcar of width $4.3\,\mathrm{km}$ to account for the vertical resolution of the measurements and to reduce vertical oscillations. The used width corresponds to the approximate vertical range covered during one readout (from combination of vertical size of the IFOV and the scan). This smoothing essentially also defines the vertical resolution of the resulting trace gas profiles. Reasonable results for greenhouse gases are achieved for altitudes between about 17 and 45 km, see e.g. Noël et al. (2018). At the wavelengths considered in the present study and with the improved calibration performed here we expect that this validity range can be extended even to somewhat lower altitudes, see also below.

#### 4.1.2 Specific settings and sequence of fits

The general ONPD settings are the same as described in Noël et al. (2018). We use a vertical layering from 0 to 50 km with 1 km steps. In general, the ONPD method uses a fixed data base of reference transmissions derived with SCIATRAN assuming conditions of the 1976 US standard atmosphere (NASA, 1976). We correct for the actual conditions by using corresponding weighting functions via Eq. (4). For pressure and temperature this is done by using as input data from the ECMWF ERA

Interim model. We select the profiles spatially and temporally closest to the measurements and interpolate them to the ONPD altitude grid.

In the current study, we have performed calculations for three different aerosol extinction wavelengths $\lambda_{\text{aer}}$ (452, 525 and 750 nm). The degree of the fitted polynomial is 2 in these cases. For consistency reasons and because the fitting windows are optimised for the aerosol extinction retrieval we use a specific sequence of retrievals such that information obtained in one retrieval can be used in other retrievals. Therefore we start with the retrieval for $\lambda_{\text{aer}}$=525 nm, from which we obtain $O_3$ and $NO_2$ profiles which are then used in the other retrievals. The detailed settings for each retrieval are summarised in Table 2.

## 4.2 Aerosol extinction retrieval

The standard ONPD method does not require fully calibrated data as input because the fitted polynomials $P_j$ also account for possible multiplicative radiometric offsets, i.e. as caused by the scan over the sun.

In the present study we use fully calibrated transmissions as input. Therefore, the polynomials $P_j$ should essentially contain information about aerosol extinction in the atmosphere. This can be described by the following formula:

$$P_j(\lambda_{\text{aer}}) = -\sum_i \epsilon_i(\lambda_{\text{aer}}) \, l_{ij}(\lambda_{\text{aer}}) \tag{5}$$

$P_j(\lambda_{\text{aer}})$ is the value of the polynomial $P_J$ derived from the ONPD retrieval at the wavelength $\lambda_{\text{aer}}$, at which we want to determine the aerosol extinction. $l_{ij}$ is a (fixed) geometric factor which describes the length of the occultation light path in layer $i$ when looking layer $j$. These path lengths are also derived from SCIATRAN for each atmospheric layer and viewing direction and consider refraction. They therefore also depend slightly on wavelength. $\epsilon_i$ is the aerosol extinction in layer $i$; this is the quantity we want to derive.

This is done – consistently with the ONPD approach – by use of an onion peeling method: We start at the top layer and then propagate downwards while taking into account the results from above. Contributions from below the current tangent $j$ (due to refraction and vertical size of the IFOV) are considered by assuming $\epsilon_i = \epsilon_j$ for $i < j$ when determining $\epsilon_j$. Since aerosol extinction typically increases with decreasing altitude this results in a small over-estimation, but gives a stable solution.

## 5 Results

### 5.1 Example 11 September 2003

To illustrate the outcome of the different calibration and retrieval parts described in the previous section we present in this subsection as an example the results for orbit 8014 (on 11 September 2003). This orbit has been selected due to a collocation of SAGE-II and SCIAMACHY limb measurements, such that a direct comparison of aerosol extinction results is possible (see below).

Fig. 5 shows the transmissions as function of altitude for the three selected aerosol extinction wavelengths. In the left column the uncorrected transmissions (i.e. without scan correction) are shown in red (similar to the data shown in the top left graph

of Fig. 3). The black dots denote the selected subset of data which is used in the retrieval. Effects of the scan over the sun are visible.

The right column of Fig. 5 shows the selected transmissions after the corrections explained above, which now smoothly decrease with altitude as it is expected. The variation of transmission with altitude is different for each wavelength due to different absorbing and scattering effects. In general transmissions at shorter wavelengths are lower at lower altitudes mainly due to ozone absorption and stronger Rayleigh scattering. Below 10 km transmissions are close to zero due to the low input signal, which gives a lower limit for the later retrieval. At altitudes above about 30 km transmissions are close to one. Since aerosol extinction information is obtained from the difference of the transmission to one, this also implies an upper limit for the retrieval (see below).

The selected and corrected spectra are then fed into the ONPD retrieval (see section 4.1), in which the background polynomial is fitted considering gas absorptions and Rayleigh scattering. The results of this retrieval for orbit 8014 are shown in Fig. 6. The left column of this figures shows (again for the different aerosol extinction wavelengths):

– The corrected measured logarithmic transmission at 25 km (thick grey line).

– The SCIATRAN reference model spectrum for US standard atmosphere conditions, incl. Rayleigh scattering (green line).

– The model spectrum corrected for actual temperature, pressure and absorption of gases as derived from the fit (blue line).

– The fitted background polynomial (pink line).

– The fitted spectrum, i.e. the combination of the contributions of reference spectrum, absorption and polynomial (red line).

As the fit result (red) is very close to the measurement (grey) the right column of Fig. 6 shows the residual of both (measurement – fit), which is quite low (standard deviation below 0.002) indicating a good fit.

The white circles on the pink lines in Fig. 6 mark the value of the polynomial at the wavelength to be used for aerosol extinction retrieval. This is the value for 25 km; the complete profiles from 10 to 50 km are presented in the top of Fig. 7. These profiles show the remaining transmission after effects of Rayleigh scattering and gas absorption have been subtracted. The difference to one can thus be interpreted as the effect of aerosol extinction.

The profiles of Fig. 7 are used as input for the aerosol extinction retrieval (see section 4.2). The resulting aerosol extinction profiles are given in Fig. 8. For comparison, we also plotted collocated SAGE-II (at 452 and 525 nm) and SCIAMACHY limb aerosol extinction (at 750 nm) profiles. In this case, the latitude/longitude of the SCIAMACHY measurement are 61.9°N/61.6°W; the SAGE-II measurements took place about 535 km/40 min apart from this. The SCIAMACHY limb measurement has a distance of 267 km/-6.6 h.

The error bars correspond to the error given in the product files. For SCIAMACHY occultation, this error is derived from the propagation of the transmission errors (Fig. 7 bottom). It does not consider any systematic contributions and is therefore only a lower estimate.

The overall agreement between SCIAMACHY occultation and SAGE-II is quite good. Above about 30 km transmissions are close to one (see Fig. 7). Thus, SCIAMACHY occultation errors typically increase and the retrieved aerosol extinctions become very noisy. Furthermore, at higher altitudes vertical oscillations occur, which are artefacts probably introduced by the onion peeling method; similar effects have been seen in greenhouse gas retrievals (see e.g. Noël et al., 2018).

At 750 nm, the retrieved SCIAMACHY limb and aerosol extinctions are also quite similar. The vertical sampling of the limb data is however much sparser. Noise and error of the occultation data is smaller; oscillations at higher altitudes are more pronounced than at lower wavelengths. The aerosol extinction minimum in the limb data at about 15 km is not seen in the occultation data.

## 5.2 Validation

In this section we show the results of a comparison between the SCIAMACHY solar occultation V5.1.1 aerosol extinction data and corresponding profiles from other sensors, namely solar occultation data from SAGE-II and SAGE-III and limb profiles from SCIAMACHY and OSIRIS. For the comparisons, all aerosol extinction data are interpolated to the 1 km SCIAMACHY occultation data vertical grid. Only altitudes which are valid in both data sets are used; if not explicitly mentioned below, no additional filtering is applied. We compute for all collocated data of each data set mean profiles and corresponding standard deviations. Then, we determine at each altitude the mean difference (SCIAMACHY – reference), the corresponding standard deviation of the difference and the mean of the error given in the SCIAMACHY product. These values are then divided by the mean aerosol extinction of both data sets to give relative values.

Because of the larger random and/or systematic errors at higher altitudes (see previous subsection) we currently consider only SCIAMACHY solar occultation aerosol extinction data below 30 km as reliable. In addition, SCIAMACHY occultation data below about 15 km have to be treated with care, as e.g. the greenhouse gas occultation retrievals are known to give less accurate results there because of tropospheric influences not covered by the retrieval method (like increased refraction and strong vertical gradients at the tropopause). For the validation activities described in this section and later analyses we will therefore concentrate on the altitude range 15–30 km.

### 5.2.1 Comparison with SAGE-II

The results from the comparison with SAGE-II at 452 and 525 nm are shown in Fig. 9.

Since aerosol extinctions exponentially decrease with altitude, mean differences and standard deviations of the differences decrease towards higher altitudes whereas relative differences increase. In general, there is no obvious bias between the SCIA-MACHY occultation results and the correlative data sets visible, but especially at 452 nm the mean occultation profile shows an oscillation with altitude which is not present in the SAGE-II data. This results in an oscillation of the differences with an amplitude of about 20–30% and an estimated period of about 10 km. For upper altitudes (above about 25 km) at 525 nm this oscillation even causes mean differences larger than 50% to SAGE-II.

These kind of oscillating features have been observed in other ONPD products (see e.g. Noël et al., 2018). It is assumed that these are related to the onion peeling method which does not include e.g. regularisation on these vertical scales.

The mean error of the SCIAMACHY occultation product is at all wavelengths smaller than the standard deviation of the differences confirming that this error is indeed only a lower estimate. The standard deviation of the mean profiles is very similar for all comparisons. This indicates that all instruments / viewing geometries observe a comparable atmospheric variability.

### 5.2.2 Comparison with SAGE-III

The SAGE-III instruments on Meteor-3M provides aerosol extinction profiles at wavelengths close to those of the SCIA-MACHY solar occultation product which allows a direct comparison for all three wavelengths (see Fig. 10) with significantly more (5505) collocations than for SAGE-II for a similar time interval (2002 to 2005).

The results around 450 nm are very close to those obtained when comparing with SAGE-II. Between about 17 and 27 km SCIAMACHY and SAGE-III data agree within about 20%. Above and below deviations are larger (up to 60% at 30 km, with SAGE-III aerosol extinctions being larger that those of SCIAMACHY. The oscillation features are also clearly visible.

Around 525 nm the agreement with SAGE-III is very good below about 24 km; deviations are smaller than about 10% with only small oscillation. Above this altitude, oscillations increase leading to differences of up to 50–60%.

At 750 nm there seems to be a systematic offset in the aerosol extinctions; SCIAMACHY data are about 30–50% higher. Some small vertical oscillations are also visible here. Above about 25 km deviations start to increase up to values above 100% at 30 km.

### 5.2.3 Comparison with SCIAMACHY limb data

For the comparison of SCIAMACHY occultation data with limb aerosol extinctions at 750 nm we divided the collocation data set into two parts corresponding to background conditions (defined by maximum aerosol extinctions below 0.001) and perturbed conditions (all others). The results are shown in Fig. 11.

Because of the large number of collocations the error of the mean difference is very small (dotted and solid red lines are almost on top of each other).

For the background case, the comparison reveals an almost perfect agreement between 20 and 25 km; below 20 km and up to 27 km there is a small offset of ±10–20%. Above 27 km differences start to increase reaching about 80% at 30 km. The standard deviations of the mean profiles are very similar for occultation and limb data, so variability is also comparable.

Under perturbed conditions, the atmospheric variability is much higher both in the spatial and temporal domain. The time offset of up to 10 h between occultation and limb measurements therefore results in a larger scatter between the two data sets and significantly increased standard deviations of differences and mean profiles of more than 100%. This is why the lower limit lines of the standard deviations are not always visible in the logarithmic profile plot (d). The variability for limb is even larger than for occultation, possibly because occultation measurements occur always at the same local time (sunset). However, the average agreement of the two data sets is very good between about 17 and 27 km (deviation smaller than 10%).

Below 17 km deviations up to 50% are observed. This is in line with comparisons of OSIRIS and SCIAMACHY limb aerosol extinctions with SAGE-II data (Rieger et al., 2018), which also revealed discrepancies of similar magnitude and sign at higher latitudes. It is assumed that these differences are due to the assumptions on particle sizes made in the limb retrievals,

which are most crucial for high Northern latitudes because of low scattering angles. This especially plays a role under perturbed conditions at lower altitudes, where the size distribution changes due to the insertion of volcanic particles. Perturbations in the particle amount and their sizes due to volcanic eruptions rapidly decrease with the altitude and usually do not reach above 20 km in the period from 2002 to 2012.

Above 27 km deviations increase with occultation data being typically larger. This is most likely also related to oscillations in the occultation profiles (see Fig. 8).

### 5.2.4   Comparison with OSIRIS

The results of a direct comparison with OSIRIS limb aerosol data at 750 nm shown in Fig. 12 reveal that the SCIAMACHY solar occultation aerosol extinctions are on average 20–30% larger than those from OSIRIS, again with even larger differences

above about 27 km. This supports the assumption, that the deviations at higher altitudes can be attributed to the SCIAMACHY data.

Results for disturbed and background conditions are similar in this case; this applies also to the variability (standard deviation of difference), which is on the order of 20-30% with higher values at altitudes below 17 km and above 25 km. This lower variability compared to SCIAMACHY limb data is due to the fact that the collocated OSIRIS data do not contain mea-

surements at high latitudes in winter, where atmospheric variability is strongly increased by the polar vortex. Furthermore, OSIRIS measurements at high Northern latitudes are less affected by the assumed particle size distribution as they are done at scattering angles close to 90°contrary to the forwards scattering conditions typical for SCIAMACHY limb measurements at these latitudes.

## 6   Time series

### 6.1   Aerosol extinction time series

The complete time series of SCIAMACHY solar occultation data has been processed for the three aerosol extinction wavelengths investigated in the present study. After filtering out invalid data (from times of non-nominal instrument performance, e.g. during decontamination periods) in total 43686 profiles (from August 2002 to April 2012) remained, from which daily average aerosol extinction profiles were created. Because of the sun synchronous ENVISAT orbit, all measurements of one

390    day occur at essentially the same latitude but different longitudes. Thus, the geographic latitude of the measurements varies systematically with season and the daily averages are also zonal means (see also Noël et al., 2018). Higher latitudes ($\sim$65–70°) typically occur in winter and lower latitudes ($\sim$50–60°) in summer.

Fig. 13 shows the resulting gridded time series from August 2002 to April 2012 for 452 nm, 525 nm and 750 nm (top to bottom plots) from 15 to 30 km. The colour scale is logarithmic accounting for the typical exponential decrease of the aerosol

extinction with altitude which is clearly visible in this figure at all wavelengths.

After 2008 there are some pronounced increases of aerosol extinction up to about 0.01 at lower altitudes visible. These are caused be the eruption of volcanoes (marked by arrows) which reached into the stratosphere. In the case of Nabro, the eruption occurred at low latitudes (13°N), but the plume was then transported to higher latitudes. The sudden increase due to upward transport of aerosol particles directly after the eruption is then followed by a gradually downward transport and decrease of aerosol extinction taking several months up to one year. This can be seen at all wavelengths.

The observed aerosol extinctions also vary with season, which is partly caused by the systematic coupling between time and latitude mentioned above and the related variations in tropopause height.

## 6.2 Anomalies

To further investigate the temporal behaviour and to reduce the influence of possible systematic features in the data (e.g. vertical oscillations, see above) we computed monthly relative anomalies of the aerosol extinction. We concentrate here on the years 2003 to 2011 to avoid possible influences of missing months in the first and the last year on the weighting of data points.

For this, we first generated for each altitude monthly means from the daily average profiles. From these monthly averages we then subtracted the 2003 to 2006 average value for each month to obtain absolute anomaly profiles. These data are then divided by the mean of the monthly average aerosol extinction profiles from 2003 to 2006 to remove the overall vertical shape of the aerosol extinction profiles (especially the exponential decrease with altitude). We do not use data after 2006 to determine the mean aerosol extinction profiles to avoid influences of the prominent volcanic eruptions at lower altitudes (as seen in Fig. 13). The reference for the anomalies can therefore be interpreted as a "background time" mean.

The resulting relative anomalies may then be plotted using a common linear scale for all altitudes which facilitates the interpretation of the data. As already seen in the aerosol extinction plots (Fig. 13) aerosol extinctions increased during times of volcanic influences by more than a factor of 10. These events are of course also clearly visible in the relative anomalies, but here we want to focus on smaller effects which cannot directly be inferred from the aerosol extinction time series. Therefore we concentrate on the range of relative anomalies between $\pm 4$. Fig. 14 shows the monthly relative anomalies generated by the procedure described above using this scale.

Below 20 km, in addition to the three periods of volcanic influences after mid 2008 the "background time" before 2007 can be clearly identified. During this time interval relative anomalies are close to zero, but slightly increasing with time.

Especially at the lower wavelengths a small increase of relative aerosol extinction anomaly at the beginning of 2007 is observed. This is related to the influence of volcanic eruptions in the tropical region in 2006 (Soufrière Hills, Tavurvur) and later transport of particles to higher latitudes (see e.g. von Savigny et al., 2015).

The 525 nm data show during January 2007 an oscillating structure between 16 and 19 km. In fact, this feature is visible with different strength at all wavelengths. It is most likely induced by the presence of strong Polar Stratospheric Clouds (PSCs) partly blocking the measured signal below 20 km, which is supported by the time and location of the measurements (high latitudes in winter) and ECMWF data showing during this month at these altitudes temperatures below 195 K at which PSCs can be formed. In fact, the 750 nm plot shows several enhancements in winter time (e.g. in January 2008, 2010 and 2011) between 20 and 30 km which we attribute to (in these cases less strong) PSCs. The occurrence of PSCs at altitudes up to

430 30 km in January 2011 is quite unusual. It is confirmed also by CALIOP and MIPAS measurements and related to specific meteorological conditions during this winter leading e.g. to a record ozone loss (see e.g. Arnone et al., 2012; Khosrawi et al., 2016; Pitts et al., 2018).

Altitudes above 25 km show a regular pattern of alternating positive and negative anomalies with a period of about two years. This temporal variation is a transport effect assumed to be associated with the Quasi-Biennial-Oscillation (QBO), see

e.g. Baldwin et al. (2001).

Similar features can be seen in Fig. 15, which shows relative anomalies of the SCIAMACHY limb aerosol extinction data as function of altitude and time. This plot is based on the complete set of about 40000 collocated limb profiles used in the comparisons above (see section 5.2). These limb data have been processed the same way as the occultation data to yield the anomalies.

The results for the limb aerosol extinctions are indeed very similar to occultation, but because of the larger variability of the limb data PSCs are more frequently visible in winter time. Anomalies are also somewhat larger for volcanic events, which is in line with the validation results.

To further illustrate this, Fig. 16 shows corresponding time series at 15, 20 and 25 km together with Singapore monthly mean zonal wind data (Freie Universität Berlin, 2014), which are a proxy for QBO. Whereas the overall temporal behaviour

of the limb and occultation data sets is very similar, individual events (PSCs, volcanic eruptions) are sometimes differently pronounced due to different measurement times and locations. The correlation with the zonal winds is also clearly visible. Note that there is a temporal shift between the time series which is related to the time required to transport air from the tropics (where winds are measured) to higher latitudes which may take – depending on altitude – up to 8 years at 30 km (Haenel et al., 2015). A more detailed discussion on QBO and related transport effects (which are similar for trace gases and aerosols) is e.g.

given in Noël et al. (2018). Related results for SCIAMACHY limb data are shown in Brinkhoff et al. (2015).

## 7 Conclusions

Based on an improved radiometric calibration of SCIAMACHY solar occultation measurements and a newly developed onion peeling retrieval method a stratospheric aerosol extinction profiles data set at 452 nm, 525 nm and 750 nm for the time interval August 2002 to April 2012 could be derived. This data set covers the latitudinal region between about 50°N and 70°N at a

455 specific spatial/temporal sampling. Reasonable results are obtained between 15 km and 30 km.

Comparisons with SAGE-II data products at 452 nm, 525 nm show a good agreement with essentially no mean bias but altitude dependent differences on the order of 20–30%. These differences are mainly due to unexpected vertical oscillations in the SCIAMACHY aerosol extinction profiles with a period of about 10 km. It is assumed that these oscillations are caused by the onion peeling retrieval method, as similar effects have been seen in the analysis of greenhouse gas profiles derived from

460 SCIAMACHY solar occultation measurements (Noël et al., 2018). These findings are in principle confirmed by comparisons with SAGE-III data.

At 750 nm the results are less conclusive. The overall agreement with SCIAMACHY limb data at 750 nm is quite good between about 17 and 27 km (5–10%). At higher and lower altitudes deviations up to about 50% are observed, which are caused by oscillations in the occultation data (above 27 km) and deficiencies of the limb data at higher latitudes (below 17 km). The scatter in the data is especially large during perturbed / high aerosol load conditions. Corresponding OSIRIS limb data show a similar behaviour, but are typically about 25% lower than SCIAMACHY data. An even higher offset of up to 50% is derived for SAGE-III.

The observed oscillations become less prominent (amplitudes < 10%) when comparing with data sets where larger number of collocations are available covering longer times (e.g. SCIAMACHY limb and OSIRIS). They can be essentially removed by computation of anomalies.

Time series of SCIAMACHY solar occultation aerosol extinctions and related anomalies show the expected influences of major volcanic eruptions reaching the stratosphere, which cause a sudden increase of aerosol extinction by one order of magnitude or more below 20 km followed by a gradually decrease / downward transport over several months. Furthermore, some enhanced aerosol extinctions during polar winter time were detected between 20 and 30 km which are attributed to the presence of PSCs.

A systematic variation of aerosol extinctions with season is observed, which is caused by the spatial/temporal coupling of the SCIAMACHY solar occultation measurements resulting in a regular variation of the tropopause height over the year. At altitudes above 25 km also QBO effects are seen, which is in line with the results of greenhouse gas studies (Noël et al., 2018).

These results show, that the new SCIAMACHY solar occultation aerosol extinction data products are of reasonable quality and useful for geophysical interpretations. As for the corresponding greenhouse gas data the quality of the products seems to be mainly limited by systematic effects, especially by vertical oscillations with a period of about 10 km. This issue has been investigated for several years, but no solution could be found without significant reduction of the vertical resolution of the profiles. However, as we have shown in this study, the influence of these oscillations can be essentially removed by computation of anomalies.

**Appendix A: Azimuth correction**

Switching to the sun follower (SF) in azimuth at about 17 km tangent height may result in different azimuthal positions of the IFOV before/after the switch, resulting in a jump of the measured signal to a higher value. Azimuth mispointing may also occur due to a mismatch between the predicted (commanded) and true sun position. This is only critical, if the angular shift is so large that part of the sun is not inside IFOV (see right plot in Fig.2). The effect on the signal due to this missing area can be corrected using the known position of the IFOV on the sun (see above), but this requires the knowledge about the width of the IFOV. Unfortunately, there is not much information from SCIAMACHY on-ground calibration about the IFOV in solar occultation geometry, because this uses a smaller aperture than in the standard Earthshine measurements. This small aperture reduces the light by 3–4 orders of magnitude, which makes measurements with typical on-ground light sources difficult as

they would require long integration times. Usually, a typical value of 0.72° is given for the small aperture IFOV width (see e.g Gottwald and Bovensmann, 2011).

To investigate the impact of azimuthal jumps in the signal after switching on the SF on the final aerosol product we looked at discontinuities in the retrieved aerosol extinctions around 17 km as function of IFOV width. It turned out that only data a few kilometres around 17 km are affected by the azimuth jumps. Smoothest profiles are achieved when assuming an IFOV width of 0.68°, which is why we used this value in our study.

## Appendix B: Bending angle fit

The underlying assumption for the determination of the bending angle is that the atmosphere does not change during one occultation measurement. This, however, is a general assumption of the retrieval method. The bending angle can then be determined using the fact that altitudes of adjacent upward/downward scans overlap. This is illustrated in Fig. A1, which shows as example the (uncorrected) measured transmissions $T_1^{\mathrm{m}}$ and $T_2^{\mathrm{m}}$ of two upward scans (nos. 18 and 20). These two measurements are centred around different tangent heights, but the covered altitude ranges overlap. Let $P_1$ be the point where the transmission of scan 18 is highest. This occurs at a tangent altitude $z_1$ of about 33.5 km. If we interpolate the transmissions of scan 20 to this altitude, we get point $P_2$. The points ($P_1$,$P_2$) therefore correspond to an observation of the same tangent altitude, but for different viewing directions ($\gamma_1$, $\gamma_2$) and for different sun positions ($\beta_1$, $\beta_2$). Since the observed point in the atmosphere is the same, the scan-corrected transmissions should also be the same, i.e.:

$$T_1(z_1) = T_2(z_1) \tag{B1}$$

The fact that we observe different transmissions ($T_1^{\mathrm{m}}(z_1) > T_2^{\mathrm{m}}(z_1)$) is due to refraction, i.e. the (same) bending angle $\delta(z_1)$ at this altitude.

Combining Eqs. (B1) and (3) leads to:

$$\frac{T_1^{\mathrm{m}}(z_1)}{T_2^{\mathrm{m}}(z_1)} = \frac{S(\gamma_1 - \beta_1 - \delta(z_1))}{S(\gamma_2 - \beta_2 - \delta(z_1))} \tag{B2}$$

This equation can be solved numerically to derive $\delta(z_1)$. In principle, this procedure can be applied to all pairs of scans; however, it is practically limited by the low transmissions at lower altitudes and too small refraction at higher altitudes. We therefore restrict the application to the altitude range 15 to 35 km, which gives us about five data points of $\delta$ for different tangent altitudes $z$.

We then fit a straight line to $\log \delta(z)$ to derive the parameters $a$ and $b$ from Eq. 1. This is done independently for each considered wavelength. An example for this is show in Fig. A2.

*Data availability.* The SCIAMACHY solar occultation aerosol extinction data presented in this work (V5.1.1) are available on request from the authors.

*Author contributions.*  SN developed the calibration and retrieval methods, generated the SCIAMACHY occultation aerosol extinction data set and performed the analysis of the data. KB provided the pointing corrections for the data. AR and EM produced the SCIAMACHY limb
aerosol extinction data set. All authors (incl. HB and JPB) contributed to the preparation of the manuscript.

*Competing interests.*  The authors declare that they have no conflict of interest.

*Acknowledgements.*  SCIAMACHY data have been provided by ESA. We thank the European Center for Medium Range Weather Forecasts (ECMWF) for providing us with analysed meteorological fields. SAGE-II data were obtained from the NASA Langley Research Center Atmospheric Science Data Center. This work has been funded by DLR-Bonn (SADOS-III project), by ESA (SQWG-III project), by the
530 University of Bremen and partially by DFG through the research unit VolImpact.

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

**Table 1.** Satellite measurements of stratospheric aerosols.

| Instrument | Platform | Measurement Time | Viewing Geometry | Latitude Range |
|---|---|---|---|---|
| SAM | Apollo-Soyuz | 1975 | solar occultation | proof of concept mission |
| SAM-II | Nimbus-7 | 1978 – 1993 | solar occultation | $64°$S – $80°$S; $64°$N – $80°$N |
| SAGE-I | AEM-B | 1979 – 1981 | solar occultation | $72°$S – $72°$N |
| SAGE-II | ERBS | 1984 – 2005 | solar occultation | $80°$S – $80°$N |
| SAGE-III | Meteor-3M | 2001 – 2006 | solar & lunar occultation | $30°$S – $50°$S; $50°$N – $80°$N |
| HALOE | UARS | 1991 – 2001 | solar occultation | $80°$S – $80°$N |
| POAM-II | SPOT-3 | 1993 – 1996 | solar occultation | $63°$S – $88°$S; $55°$N – $71°$N |
| POAM-III | SPOT-4 | 1998 – 2005 | solar occultation | $63°$S – $88°$S; $55°$N – $71°$N |
| GOMOS | ENVISAT | 2002 – 2012 | stellar occultation | global |
| MIPAS | ENVISAT | 2002 – 2012 | limb | global |
| SCIAMACHY | ENVISAT | 2002 – 2012 | nadir[a], limb, solar & lunar[a] occultation | global (limb); $49°$N – $69°$N (solar occ.) |
| OSIRIS | Odin | since 2001 | limb | global |
| ACE-MAESTRO | SCISAT | since 2003 | solar occultation | global |
| CALIOP | CALIPSO | since 2006 | nadir | $82°$S – $82°$N |
| OMPS | Suomi NPP | since 2011 | nadir[a] & limb | global |
| SAGE-III | ISS | since 2017 | solar occultation | $60°$S – $60°$N |

[a] no stratospheric aerosol data

**Table 2.** Sequence and settings of ONPD retrieval.

| Sequence No. | $\lambda_{\mathrm{aer}}$ | Fit interval | | Considered absorbers (source/fit) | | |
|:---:|:---:|:---:|:---:|:---:|:---:|:---:|
| 1 | 525 nm | 510 – 580 nm | pressure (ECMWF) | temperature (ECMWF) | $O_3$ (Fit) | $NO_2$ (Fit) |
| 2 | 452 nm | 440 – 460 nm | pressure (ECMWF) | temperature (ECMWF) | $O_3$ (525 nm) | $NO_2$ (525 nm) |
| 3 | 750 nm | 750 – 758 nm | pressure (ECMWF) | temperature (ECMWF) | $O_3$ (525 nm) | |

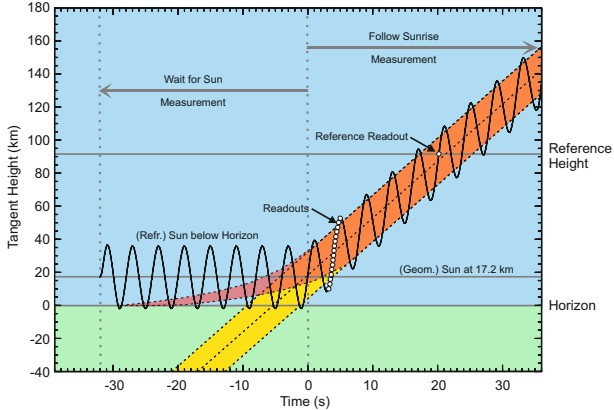

**Figure 1.** SCIAMACHY solar occultation measurement (figure from Noël et al., 2016).

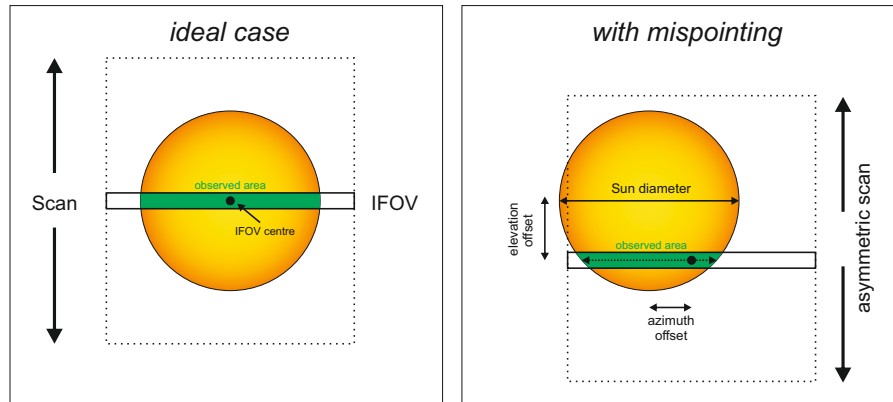

**Figure 2.** SCIAMACHY instantaneous field of view (IFOV) while scanning over the sun. Left: Ideal case (sun in centre of IFOV). Right: With mispointing (shift between centres of sun and IFOV); this is actually the nominal case.

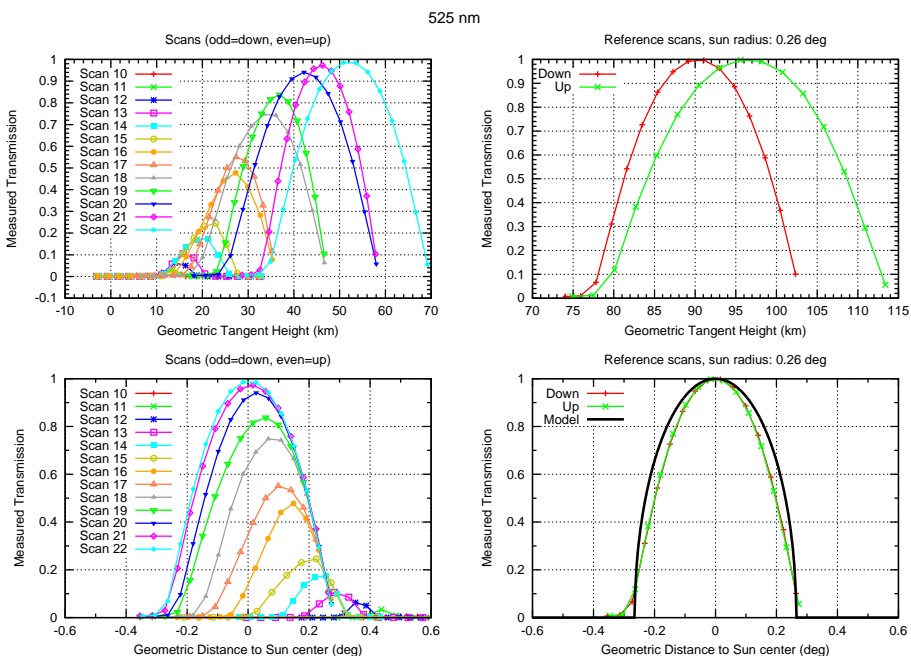

**Figure 3.** SCIAMACHY transmissions at 525 nm. Top left: Measured transmission as function of tangent altitude for different scans. Top right: Reference transmissions as function of tangent altitude for up- and downscan. Bottom left: Measured transmission as function of distance to sun centre. Bottom right: Transmissions as function of distance to sun centre and corresponding model.

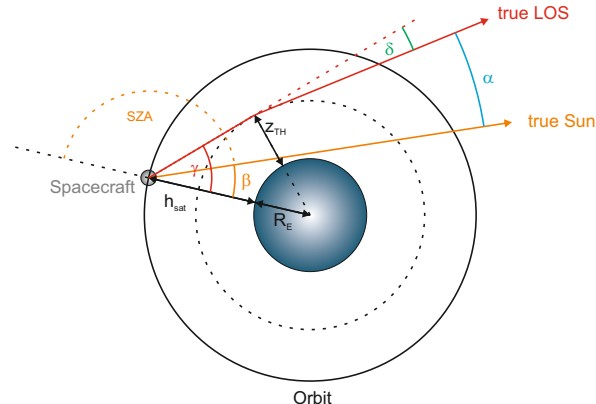

**Figure 4.** Definition of angles and other quantities used in the refraction model.

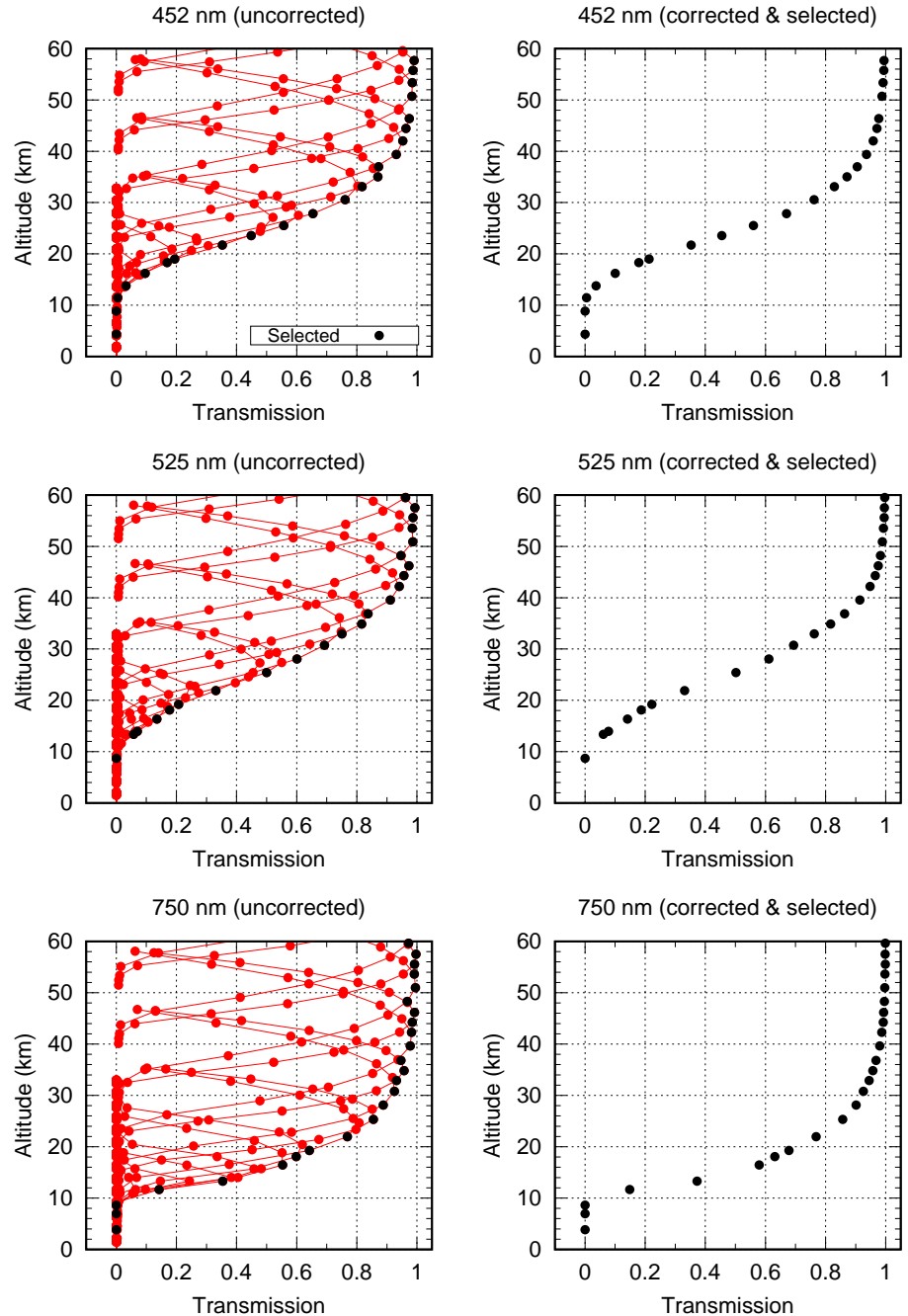

**Figure 5.** Transmissions for orbit 8014 (11 September 2003) and different wavelengths (top to bottom). Left: Uncorrected data (red) and selected sub-set (black). Right: Corrected and selected data.

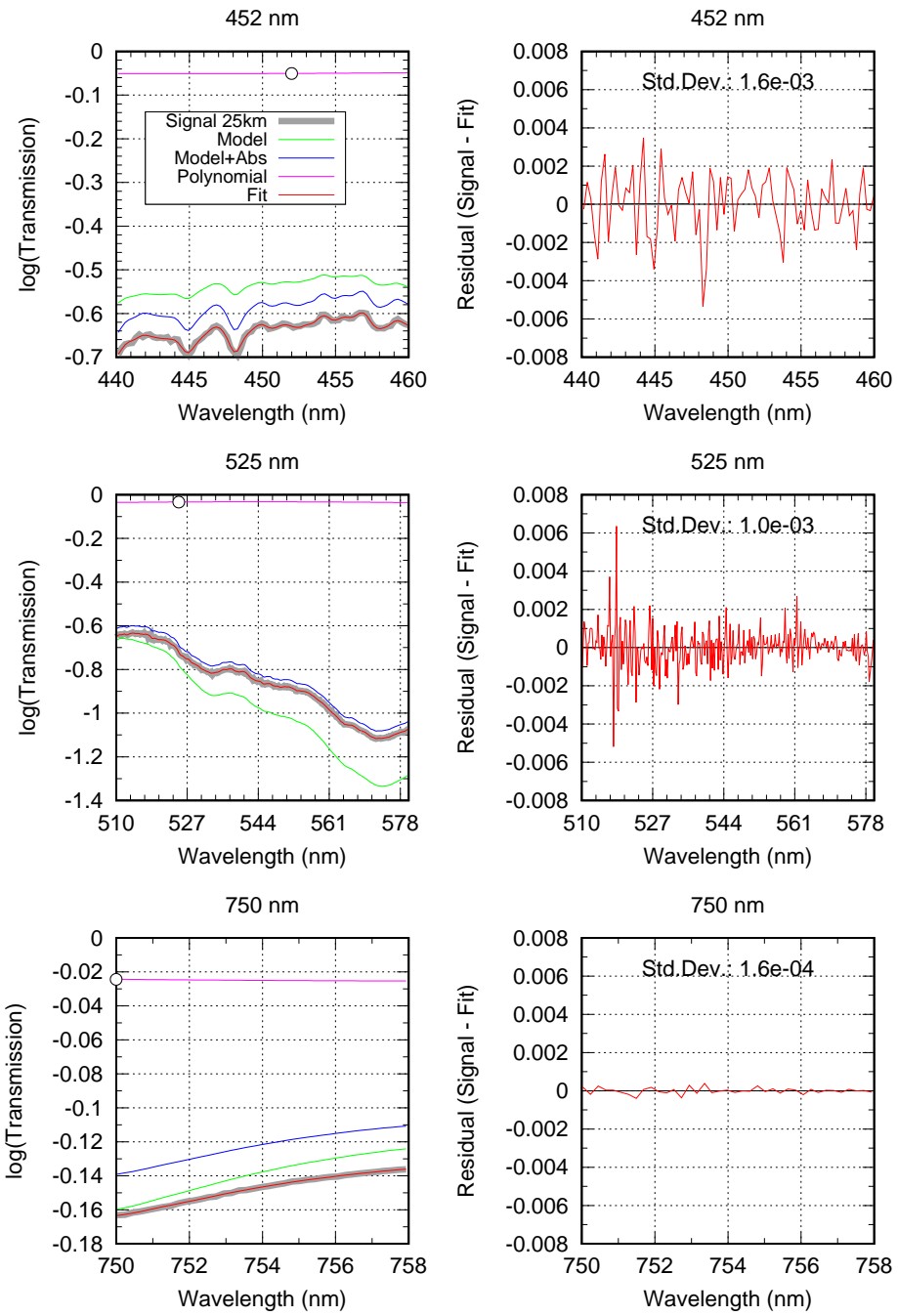

**Figure 6.** Example fit results for orbit 8014 (11 September 2003) and different wavelengths (top to bottom). Left: Spectrum at 25 km tangent height (thick grey line) and related fit results: The red line shows the total fit result, the green line the model spectrum, the blue line the model corrected for (fitted) absorptions and the pink line the (fitted) background polynomial. Right: Fit residual. The circle in the left plots marks the derived value for $P_j(\lambda_{\mathrm{aer}})$.

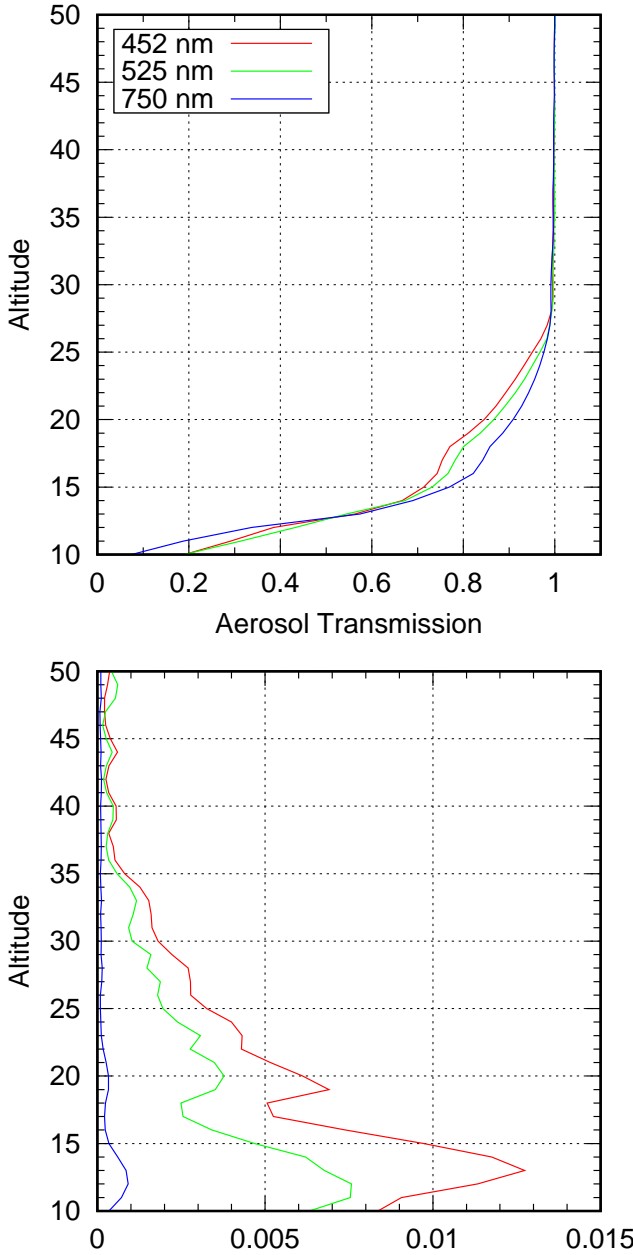

**Figure 7.** Aerosol transmissions (top) and corresponding errors (bottom) for orbit 8014 (11 September 2003) and different wavelengths.

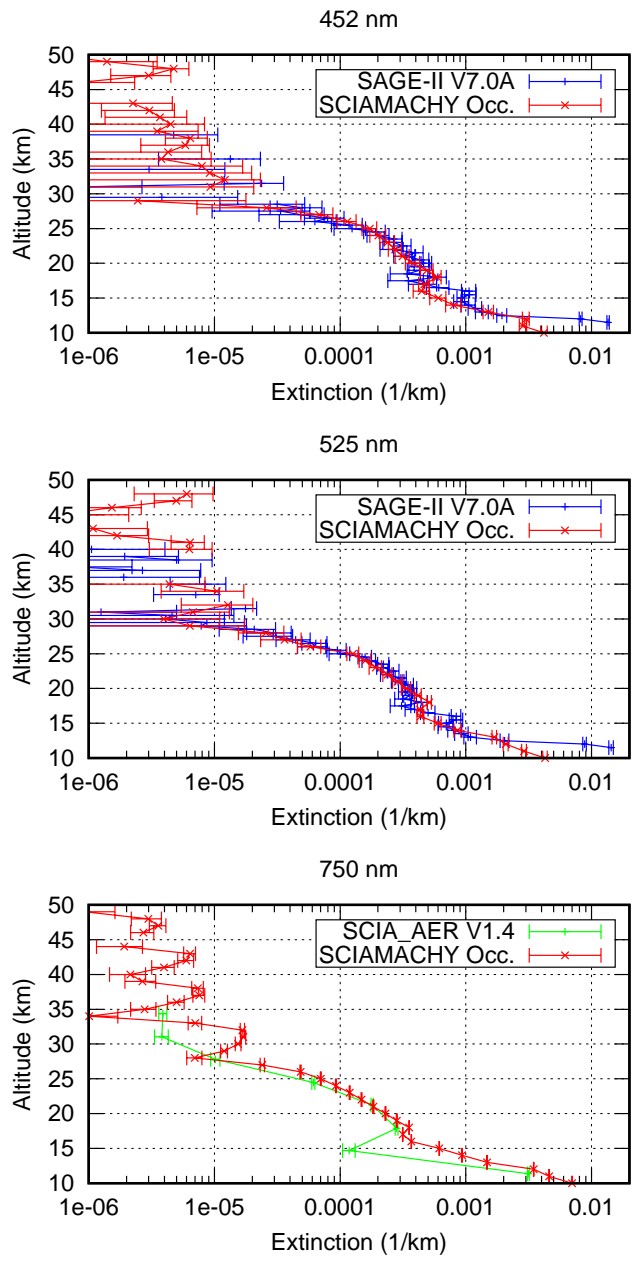

**Figure 8.** Retrieved aerosol extinction profiles from SCIAMACHY solar occultation (red), SAGE-II product V7.00A (blue) and SCIA-MACHY limb aerosol extinction product V1.4 (green) for orbit 8014 (11 September 2003) and different wavelengths (top to bottom).

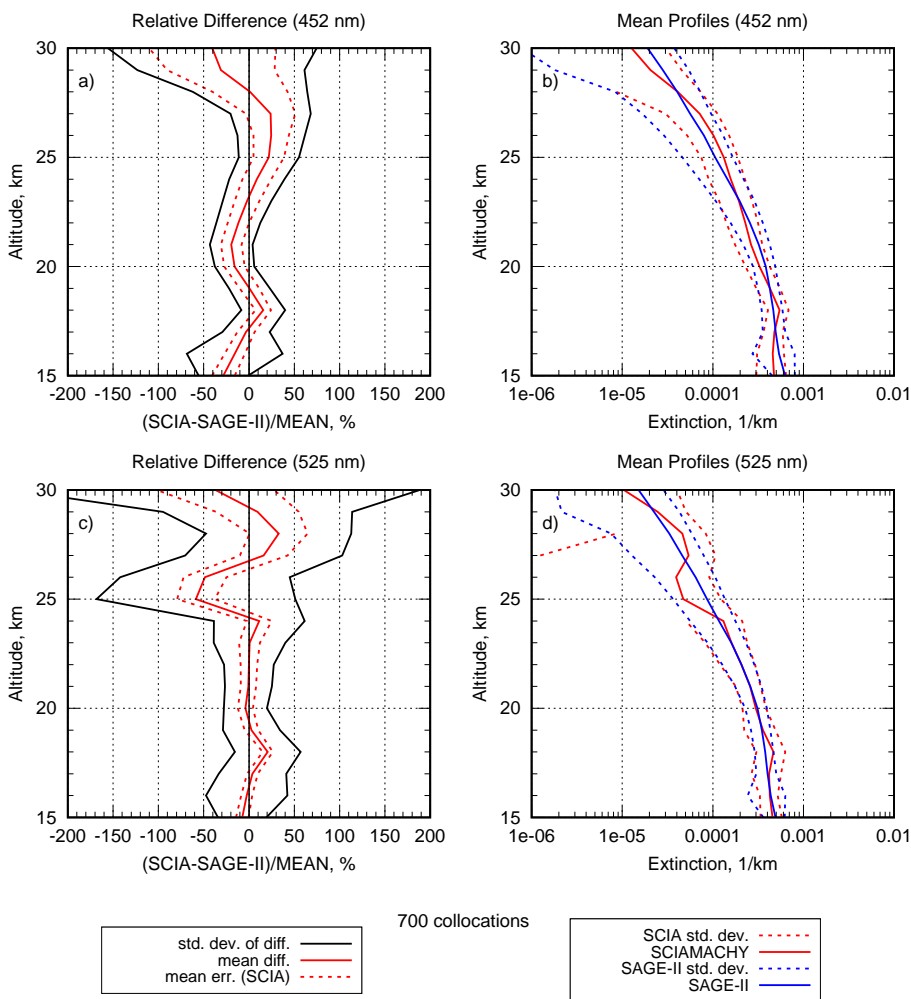

**Figure 9.** Comparison between aerosol extinction profiles from SCIAMACHY solar occultation and collocated SAGE-II data. a) Relative differences at 452 nm (solid red), mean SCIAMACHY error (dashed red) and standard deviation of differences (black). b) Mean aerosol extinction profiles at 452 nm (solid) and corresponding standard deviations (dashed) for SCIAMACHY (red) and SAGE-II (blue). c) same as a), but for 525 nm. d) same as b), but for 525 nm.

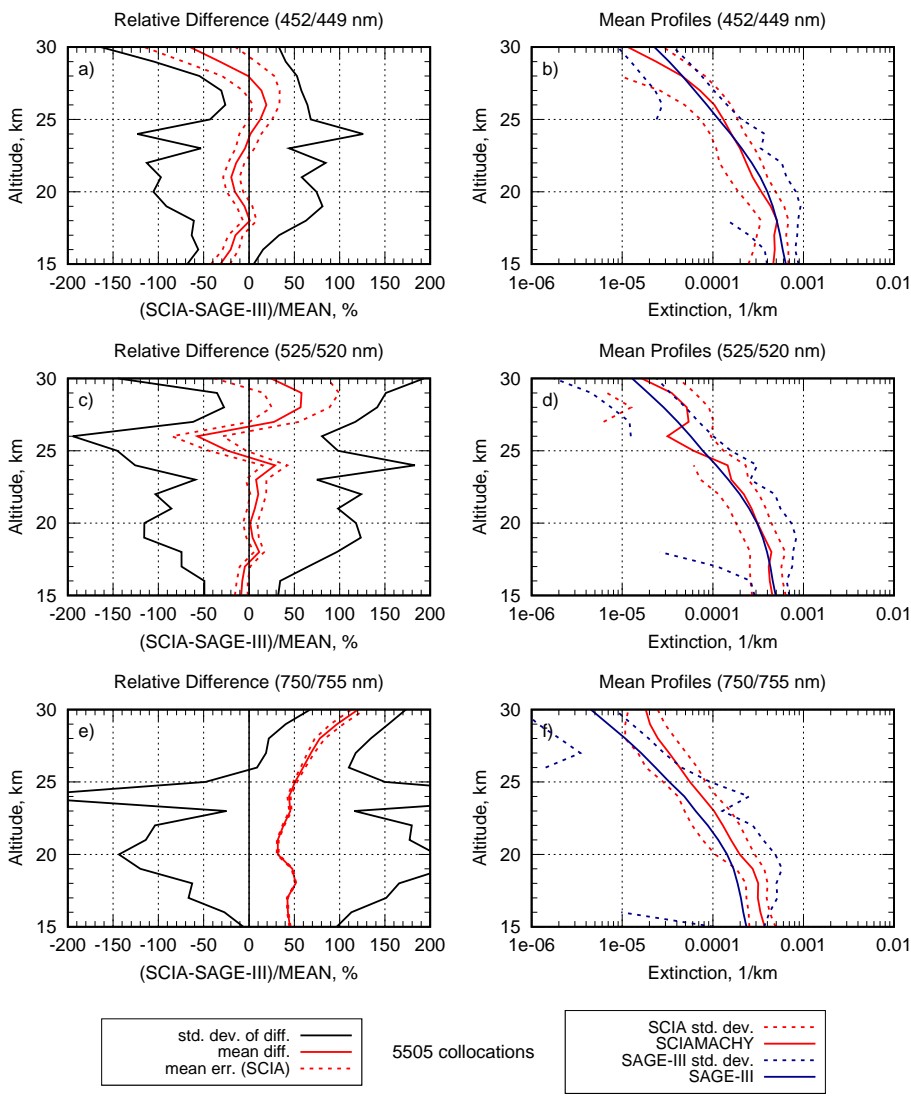

**Figure 10.** As Fig. 9, but for comparison with SAGE-III aerosol extinctions at 452/449, 525/520 and 750/755 nm (first wavelength is for SCIAMACHY, second for SAGE-III).

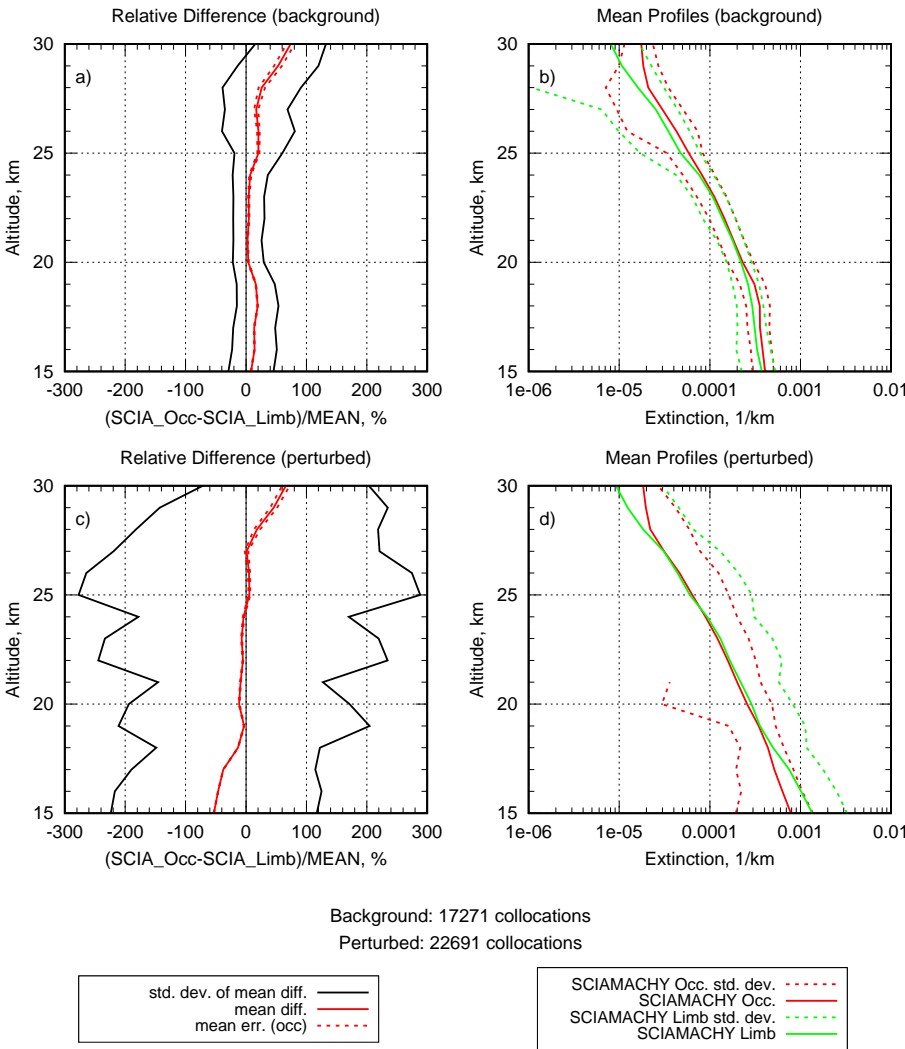

**Figure 11.** Similar as Fig. 9, but for 750 nm and comparison with SCIAMACHY limb aerosol extinctions. Top: Results for background times (i.e. aerosol extinctions always < 0.001). Bottom: Results for perturbed times (all other data).

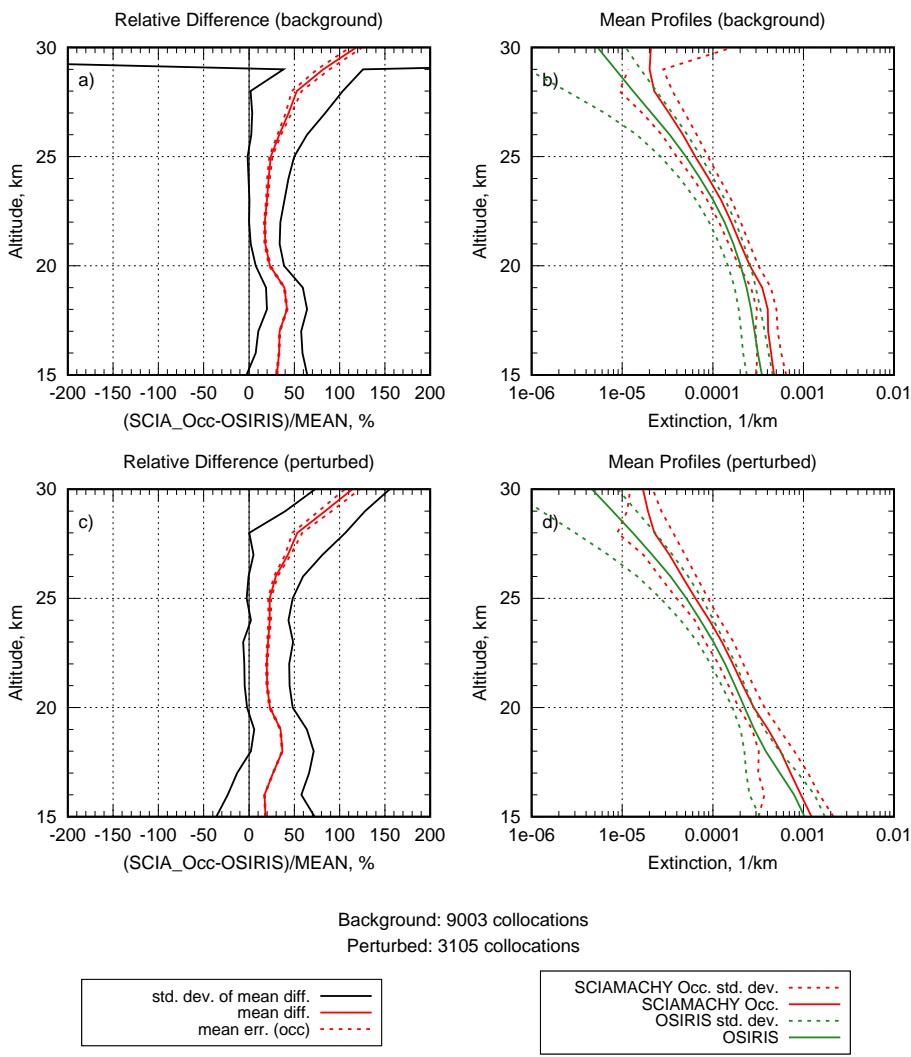

**Figure 12.** As Fig. 11, but for comparison with OSIRIS limb aerosol extinctions at 750 nm.

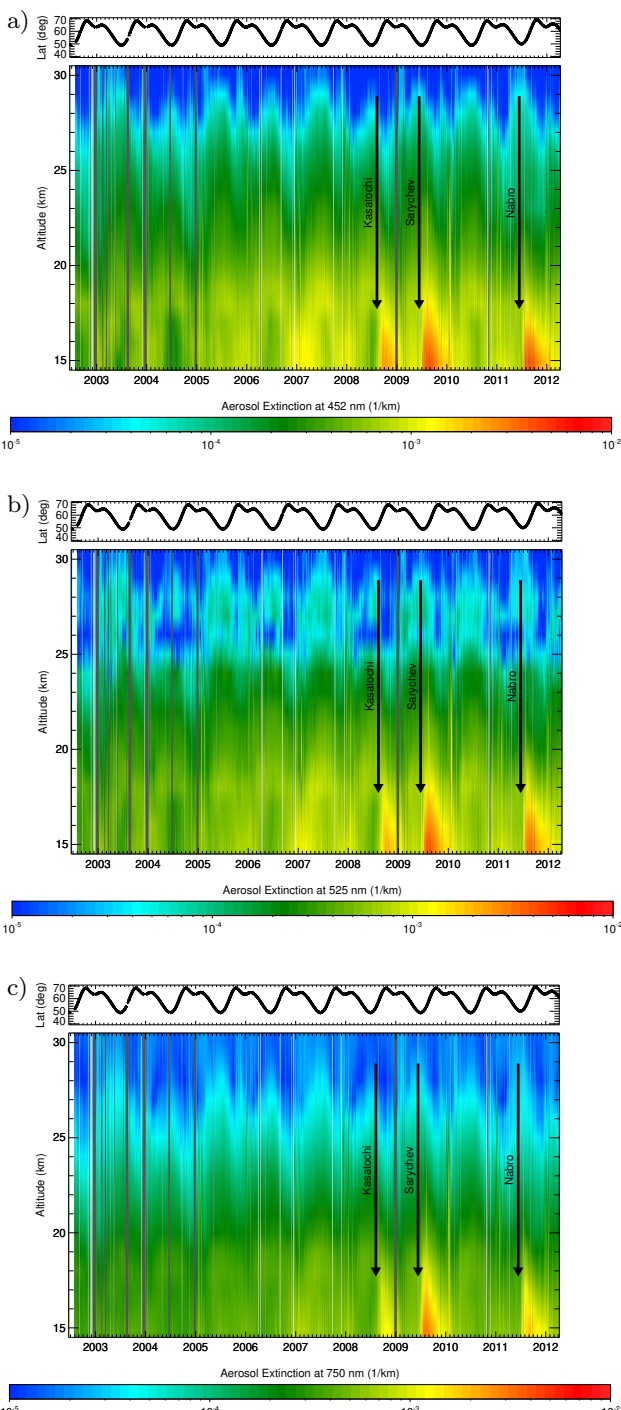

**Figure 13.** Time series of daily gridded aerosol extinction profiles from SCIAMACHY solar occultation at (from top to bottom) 452 nm, 525 nm and 750 nm. The start times of some major volcanic eruptions occurring at higher latitudes are marked. The top sub-plots show the mean latitude of the observations. Grey vertical bars denote times of degraded instrument performance (e.g. decontamination periods or switch-offs).

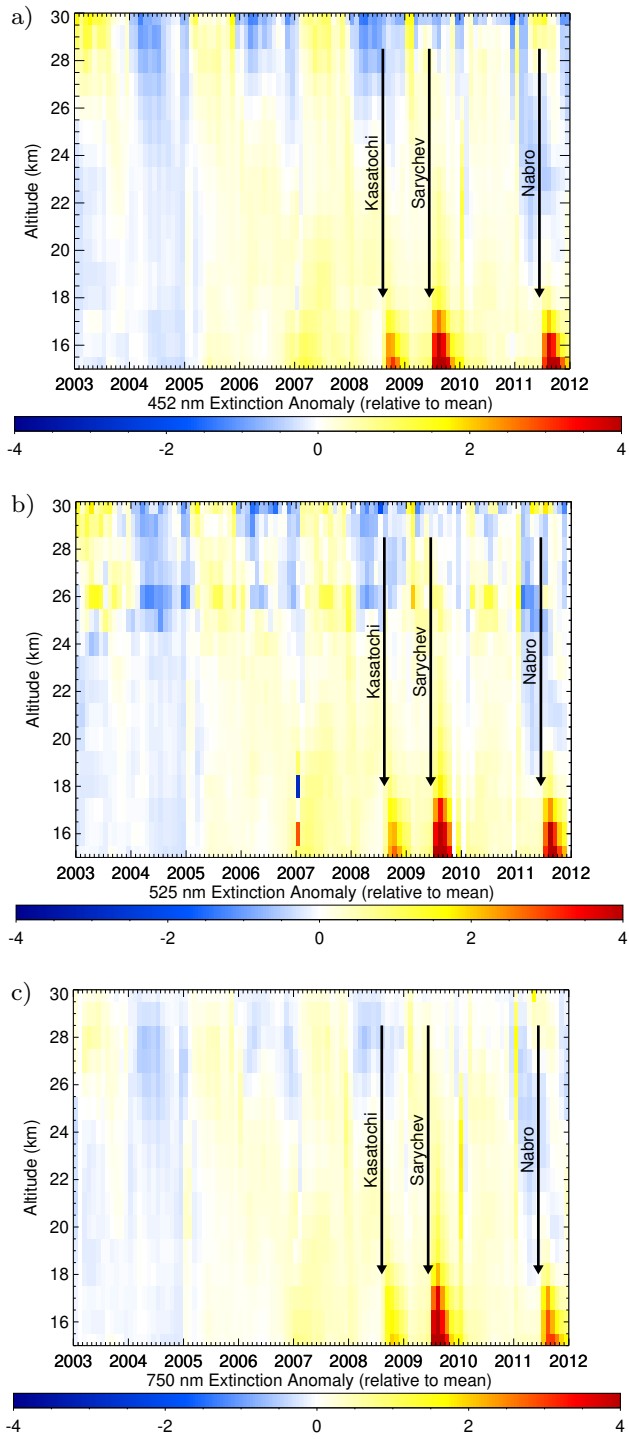

**Figure 14.** Time series of relative aerosol extinction anomalies from SCIAMACHY solar occultation at (from top to bottom) 452 nm, 525 nm and 750 nm. The start times of some major volcanic eruptions occurring at higher latitudes are marked.

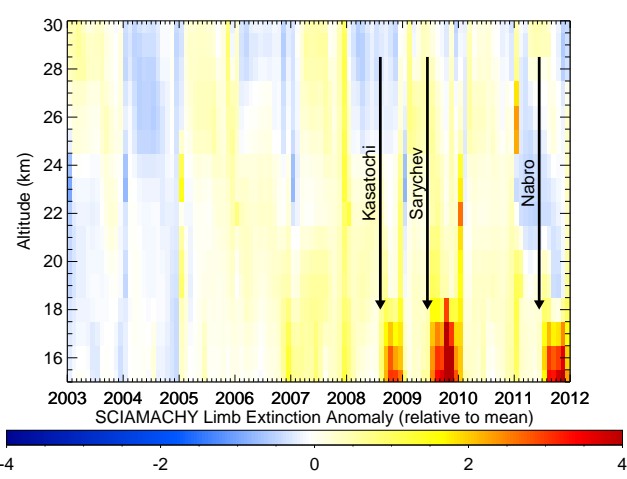

**Figure 15.** As Fig. 14, but for SCIAMACHY limb data at 750 nm.

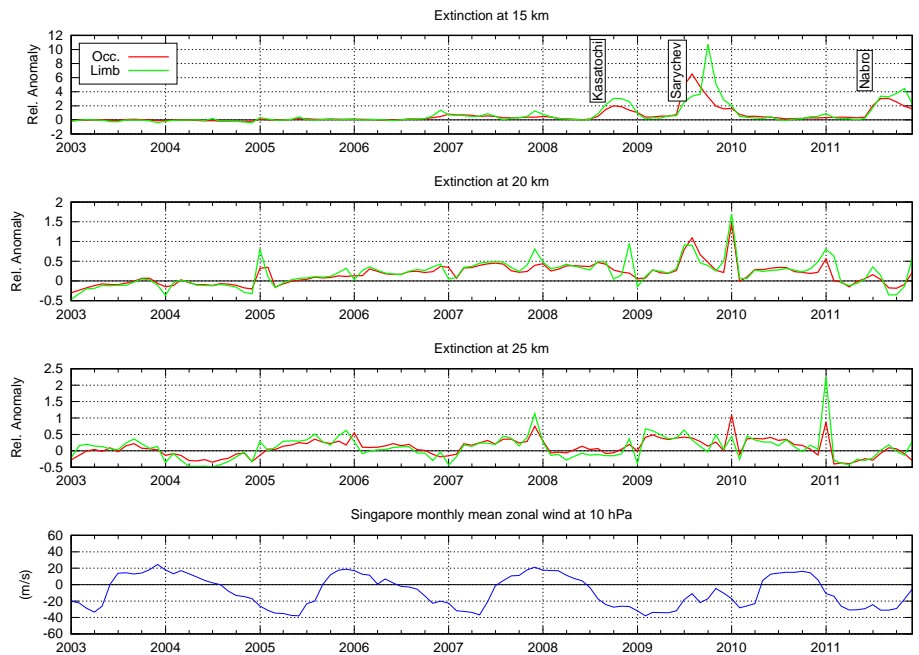

**Figure 16.** Time series of relative aerosol extinction anomalies at 750 nm for altitudes 15 km, 20 km and 25 km and Singapore monthly mean zonal wind at 10 hPa (top to bottom). Red: SCIAMACHY solar occultation data. Green: SCIAMACHY limb data. Blue: Zonal wind (as proxy for QBO).

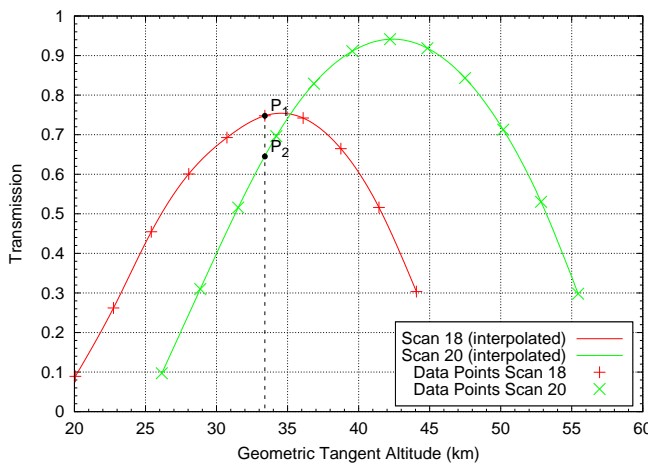

**Figure A1.** Bending angle fit. See text for explanation.

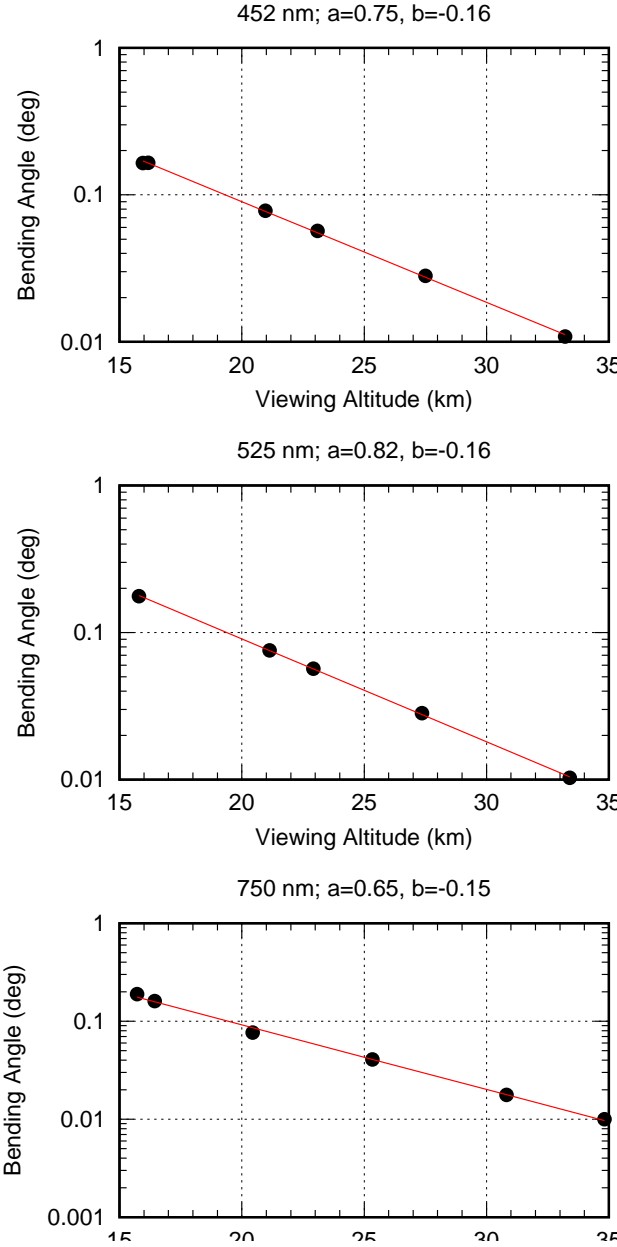

**Figure A2.** Fit of bending angle parameters $a$ and $b$ for different wavelengths.