# Peer review of "Stratospheric Aerosol Extinction Profiles from SCIAMACHY Solar Occultation"

_Atmospheric Measurement Techniques, 2020_

## Referee Comment (RC1) · Anonymous Referee #1 · 4 Jun 2020

The importance of monitoring stratospheric aerosols for climatic impacts has long been recognized and various ground-based and space based instruments have been employed for several decades. These have led to an overall understanding of background aerosols and their variability due to stratospheric dynamical processes with sporadic contributions from volcanic injections. More recently, it has been realized that, strong stratospheric aerosol loadings rivalling volcanic injections can occur from large fires in the so-called PyroCb events, thus making it even more important to continue characterizing and monitoring stratospheric aerosols. The paper by Noël et al. presents an algorithm to retrieve stratospheric extinction profiles from the solar occultation measurements by the SCIAMACHY instrument between 2002 and 2012. They have also presented initial validation of these retrievals by intercomparison with coincident occultation retrievals from SAGE II and limb scatter retrievals from SCIAMACHY. This algorithm has been used by the authors to retrieve profiles of various gas species like CO2, methane and water vapor in the past. The addition of the aerosol product should be useful for stratospheric aerosol database even though it would be of somewhat limited value because of the narrow latitude range of 50oN-70oN only. The content of the paper is well within the scope of AMT and is clearly structured. I recommend publication with some revisions. I have a few suggestions for improving the quality of the paper:

General comments:

1. As pointed out by the authors themselves, the ONPD retrieval algorithm leads to oscillations in the retrieved extinction profiles. This had been noted earlier in the retrieved profiles of gas species as well by the same authors and yet no effort has been made to ameliorate this issue. From the comparison of an individual profile with SAGE II (Fig. 8) it would appear that the oscillations are largely at altitudes over 30 km where the aerosol extinctions are very low anyway. However, later the oscillations showed up in the statistical comparison (Fig 9) at pretty much all altitudes. These oscillatory profiles make the data product of limited value. I think the paper would improve significantly by addressing this issue.

2. It will be useful to include intercomparison with some other concurrently available data products. The authors could explore using SAGE III on Meteor-3M or POAM III. In particular, the limb scatter data from OSIRIS provides good coverage spatially and temporally. The newly released level 3 stratospheric aerosol product from CALIPSO lidar also covers from ∼80oS-80oN and has good overlap in time with SCIAMACHY between 2006 and 2012. Inclusion of some of these intercomparisons will a add value to the paper.

Specific comments:

1. Page 2 line 25: The indirect effect of aerosols on the clouds may be more relevant

in the troposphere or do you mean the overshooting clouds or the cirrus clouds near the tropopause?

2. One solar occultation instrument missing in the introduction as well as in Table 1 is MAESTRO on board the Canadian SCISAT mission, e.g. see McElroy et al. 2007, Sioris et al., 2010. Also, in Table 1, please add the latitude range covered by each instrument.

3. Page 2, line 48: It is probably fair to mention clearly that CALIOP is different from the other instruments listed in Table 1 because it is an active remote sensing instrument. It is primarily intended for tropospheric aerosol extinction measurements although strato-spheric aerosol extinction retrievals have been recently produced. More relevant references for these stratospheric measurements by CALIOP are Thomason et al. (2007) and Kar et al. (2019).

4. Page 3, line 73: What do you mean by "actual" pressure and temperature profiles? In fact I am wondering why the authors used ERA-Interim rather than the newer ERA5 reanalyses. Are the pressure and temperature at mid-high latitudes in ERA-Interim better than ERA5?

5. Page 5, line 122: Please delete "exemplary" and rephrase this sentence.

6. Page 6, line 148: Please first refer to Figure 4 before this sentence.

7. Page 7, line 205: Why is 4.3 km used as the width for box car averaging? What is the impact of using a different choice on the vertical oscillation problem? Some discussion of this issue is needed here.

8. Page 9, line 240: Please mention the coincidence information between the two measurements for this case, including the latitude and longitude.

9. In Fig. 8, there is a large difference between the SCIAMACHY occultation and SAGE II profiles at the lowest altitudes (10-12 km)—could this be due to cloud related effects?

10. Page 10, line 295: For completeness, please mention how the differences with SAGE II extinction profiles were calculated in the text, although it is given in the legend to the Fig. 9. Also please mention if any filtering criteria were used.

11. Page 10, line 301: Do the results change by tightening the coincidence criteria?

12. Page 11, line 304: By "mean error", do you mean the standard error of the mean?

13. In Fig. 10, there seems to be a bias in the background case, the agreement is good mostly between 20 and 25 km with significantly larger biases above and below this altitude range.

14. Page 11 and line 320: Why do the size distribution issues affect only low altitudes―please discuss.

15. What are the black vertical lines in all the panels in Fig. 11?

16. Page 12, line 339: Note that the volcano Nabro occurred at low latitude (13oN) and the aerosol plumes spread later to higher latitudes.

17. Page 13, lines 374-376: I think the interpretation of the anomalies at altitudes above 25 km in terms of QBO is an interesting result that needs to be discussed further, rather than simply assuming it to be the case. Please add a plot of a QBO index on top of the panels in Fig. 12 so the correlation between the aerosol anomaly and the QBO can be seen more clearly and then discuss the observed anomalies at middle/high latitudes for the easterly and westerly phases of QBO and in terms of aerosol transport from the tropics. Also please discuss the effect in terms of altitude.

18. Do the linear trends shown in Fig. 15 conform to trends from other studies, if any?

19. Page 14, line 413-414: Is the QBO effect expected to be similar for gas species and aerosols?

References:

Kar, J., Lee, K.-P., Vaughan, M. A., Tackett, J. L., Trepte, C. R., Winker, D. M., Lucker, P. L., and Getzewich, B. J.: CALIPSO level 3 stratospheric aerosol profile product: version 1.00 algorithm description and initial assessment, Atmos. Meas. Tech., 12, 6173–6191, https://doi.org/10.5194/amt-12-6173-2019, 2019.

McElroy, C. T., Nowlan, C. R., Drummond, J. R., Bernath, P. F., Barton, D. V., Dufour, D. G., Midwinter, C., Hall, R. B., Ogyu, A.,Ullberg, A.,Wardle,D.I., Kar,J., Zou,J., Nichitiu,F., Boone, C. D., Walker, K. A., and Rowlands, N.: The ACE-MAESTRO instrument on SCISAT: Description, performance, and preliminary results, Appl. Optics, 46, 4341–4356, 2007.

Sioris, C. E., C. D. Boone, P. F. Bernath, J. Zou, C. T. McElroy, and C. A. McLinden (2010), Atmospheric Chemistry Experiment (ACE) observations of aerosol in the upper troposphere and lower stratosphere from the Kasatochi volcanic eruption, J. Geophys. Res., 115, D00L14, doi:10.1029/2009JD013469.

Thomason, L.W., Pitts, M.C., and Winker, D.M.: CALIPSO observations of stratospheric aerosols: a preliminary assessment, Atmos. Chem. Phys., 7, 5283–5290, https://doi.org/10.5194/acp-75283-2007, 2007.

---

## Referee Comment (RC2) · Anonymous Referee #2 · 9 Jun 2020

The authors present a new retrieval of stratospheric aerosol extinction profiles from SCIAMACHY solar occultation measurements. This has potential to be a useful data set as satellite observations of stratospheric aerosol, while important, tend to be plagued with issues due to their tenuous nature and the information poor nature of the observations. Even with the limited latitudinal sampling of the SCIAMACHY solar occultation measurements, these are valuable as the occultation approach does not have to rely on the assumptions about particle size distribution/concentration or composition that are required for lidar or limb scattering. The paper describes the retrieval approach, a preliminary "validation" of the results and some interpretation of variability in the multi-year data set. The work is well suited to AMT, however there are few issues with the paper that should be addressed before publication.

[Figure]

One of the bigger issues, which is already brought up the other reviewer, is the low frequency vertical oscillation of the profiles. This could limit the usability of the data set, and the authors make a rather sweeping assumption that this is due to the nature of the onion peeling method without regularisation. I strongly recommend further work to track down and potentially improve the oscillatory behaviour; if it is as simple as adding regularization to the retrieval, then it certainly should be done.

The other issue is the reported linear changes that are derived from the time series. The nature of the time series is highly variable due to the volcanic perturbations as nicely shown in Fig 14. The linear analysis is simply not justified. Yes, you can fit a straight line to this, but to do so is not justified, and then to report a "significant positive change of 20-30% per year" is somewhat misleading. Here the comparison with the SCIAMACHY limb scattering retrievals is quite interesting and reasonable, but with differences that the authors claim are due to different measurement times and locations. It would be better to put effort into understanding these differences and skip the linear analysis.

Other smaller issues:

Every time that an agreement between observations is claimed to be "good", please be sure to quantify.

Several times in the paper, the authors refer to the "extinction", where usually it is referring to the aerosol extinction. Please include this each time as extinction is a generalized quantity in radiative transfer and does not just refer to aerosol.

The statement at the bottom of page 2 that occultation measures extinction "whereas" limb sounders are more sensitive to smaller particles need qualification. Please explain in more detail. Do you mean limb scattering? In general limb scattering is definitely sensitive to large particles.

For SAGE II comparisons, why include the time criteria of 9 h if it doesn't matter? Also,

"temporal distance" is not a standard phrase; "time difference" is clearer.

The explanation of figure 3 needs to be clarified on page 5.

What is the numerical sun shape function, S? For a localized aerosol or cloud layer, the transmission will have a perturbed shape. How would this be handled by the algorithm to derive the shape function?

What is the impact of choosing second order for the polynomial in the fit to the transmission spectra?

Table 1 lists SCIAMACHY and OMPS nadir modes. These are not used for stratospheric aerosol to my knowledge.

Figure 3 caption uses the word "spectra" for the figure. These are not spectra.

Figure 6 caption should explain the terms in the fit.

––––––––––––––––––––––––––––––––––––

---

## Author Comment (AC1) · 31 Jul 2020

**Reply to referee 1**

We thank the referee for the detailed review. The comments will be considered in the revised version of the paper. In the following, the original reviewer comments are given in *italics*, our answer in normal font and the proposed updated text for the revised version of the manuscript in **bold** font.

[Figure]

Answers to General comments:

1. *As pointed out by the authors themselves, the ONPD retrieval algorithm leads to oscillations in the retrieved extinction profiles. This had been noted earlier in the retrieved profiles of gas species as well by the same authors and yet no effort has been made to ameliorate this issue. From the comparison of an individual profile with SAGE II (Fig. 8) it would appear that the oscillations are largely at altitudes over 30 km where the aerosol extinctions are very low anyway. However, later the oscillations showed up in the statistical comparison (Fig 9) at pretty much all altitudes. These oscillatory profiles make the data product of limited value. I think the paper would improve significantly by addressing this issue.*

    We agree with both referees that the vertical oscillations are the most critical issue for the SCIAMACHY solar occultation data product. This is why we explicitly mention it e.g. in the conclusions. These oscillations are not only a problem for the extinction retrieval but also for the greenhouse gas profile retrievals published in earlier studies. We have investigated this issue for several years, but could not identify the reasons for these oscillations. We assume they are caused by a deficiency in the radiometric calibration in combination with the onion peeling method as they seem to appear at all wavelengths. The only way to handle these in the current algorithm is to apply an additional vertical smoothing of the profiles, which we do for trace gas profiles using a boxcar of 4.3 km width. The value of 4.3 km is chosen, because this corresponds to the approximate vertical range of one readout (combination of size of instantaneous field of view and scan). We could choose a larger smoothing width here and/or apply additional smoothing to the extinction / transmission profiles. Since the oscillations have a period of about 10 km, we would need a smoothing width of at least this size, which would result in a data product with a very low vertical resolution (only $\sim$2 independent data points). We decided not to do this, as this can still be done by data users if required for a specific purpose.
Note that these oscillations become less prominent (amplitudes <10%) when comparing with data sets where more collocations covering longer times are available (e.g. SCIAMACHY limb and also the newly included OSIRIS data, see next point). This averaging effect indicates that sampling and statistics also play a role here.

Our solution to overcome the problem of vertical oscillations is to use anomalies for scientific studies (as we do in the paper). In these anomalies systematic effects – including the oscillations – are essentially removed while keeping the vertical resolution.

We will explicitly include this in the abstract and conclusions of the paper.

2. *It will be useful to include intercomparison with some other concurrently available data products. The authors could explore using SAGE III on Meteor-3M or POAM III. In particular, the limb scatter data from OSIRIS provides good coverage spatially and temporally. The newly released level 3 stratospheric aerosol product from CALIPSO lidar also covers from ∼80° S-80° N and has good overlap in time with SCIAMACHY between 2006 and 2012. Inclusion of some of these intercomparisons will a add value to the paper.*

Thank you for the suggestions. We will include comparisons with SAGE-III and OSIRIS in the paper.

Answers to Specific comments:

1. *Page 2 line 25: The indirect effect of aerosols on the clouds may be more relevant in the troposphere or do you mean the overshooting clouds or the cirrus clouds near the tropopause?*

This was a more general statement. We agree that the indirect effect is especially important in the troposphere. In the stratosphere, the indirect effect is more

related to generation of e.g. PSCs, which is mentioned in the following sentence. To clarify this, we will reformulate this part as follows:

**Stratospheric aerosols play a important role in climate as they affect radiative forcing either by scattering and absorption of light (direct effect) or by their impact on clouds and ozone (indirect effect). Especially, aerosols affect the creation of polar stratospheric clouds (PSCs) on which surfaces $O_3$ depletion takes place.**

2. *One solar occultation instrument missing in the introduction as well as in Table 1 is MAESTRO on board the Canadian SCISAT mission, e.g. see McElroy et al. 2007, Sioris et al., 2010. Also, in Table 1, please add the latitude range covered by each instrument.*

   We will add MAESTRO in the table and the text (thanks for the references). Also, latitude ranges will be included in Table 1.

3. *Page 2, line 48: It is probably fair to mention clearly that CALIOP is different from the other instruments listed in Table 1 because it is an active remote sensing instrument. It is primarily intended for tropospheric aerosol extinction measurements although stratospheric aerosol extinction retrievals have been recently produced. More relevant references for these stratospheric measurements by CALIOP are Thomason et al. (2007) and Kar et al. (2019).*

   We will mention this in the introduction and include the references.

4. *Page 3, line 73: What do you mean by "actual" pressure and temperature profiles? In fact I am wondering why the authors used ERA-Interim rather than the newer ERA5 reanalyses. Are the pressure and temperature at mid-high latitudes in ERA-Interim better than ERA5?*

   "Actual" just means that we use the pressure and temperature profiles closest to time and place of the measurement. We will clarify this in the text.
Our data product was created before ERA5 was released, therefore we use ERA-Interim data. ERA5 data have a higher spatial and temporal sampling that ERA-Interim, therefore they should indeed provide better information. However, we expect the impact of changing to ERA5 on the occultation products to be small compared e.g. to our assumption of a linear temperature correction. Especially, this would not solve the oscillation problem (see above).

5. *Page 5, line 122: Please delete "exemplary" and rephrase this sentence.*

   We will reformulate this sentence accordingly:

   **The right plots of Fig. 3 show this varying signal for the reference scan at high tangent altitudes, where atmospheric absorption and refraction are small and can be neglected.**

6. *Page 6, line 148: Please first refer to Figure 4 before this sentence.*

   Will be done.

7. *Page 7, line 205: Why is 4.3 km used as the width for box car averaging? What is the impact of using a different choice on the vertical oscillation problem? Some discussion of this issue is needed here.*

   4.3 km is the approximate vertical range covered during one readout (see above). We will clarify this in the text and add some discussion.

8. *Page 9, line 240: Please mention the coincidence information between the two measurements for this case, including the latitude and longitude.*

   Will be included.

9. *In Fig. 8, there is a large difference between the SCIAMACHY occultation and SAGE II profiles at the lowest altitudes (10–12 km) — could this be due to cloud related effects?*

As mentioned in the text, the current SCIAMACHY occultation data below about 15 km are not considered to be reliable. This is in general due to tropospheric effects, which includes clouds but also strong changes in e.g. temperature gradients which are not resolved by the instrument because of the ∼4 km resolution. One purpose of Fig. 8 is to show these limitations.

10. *Page 10, line 295: For completeness, please mention how the differences with SAGE II extinction profiles were calculated in the text, although it is given in the legend to the Fig. 9. Also please mention if any filtering criteria were used.*

We will describe in the text how the differences are calculated. We did not use specific filters, only removed those altitudes which are marked as invalid in the data (usually the lowest ones).

11. *Page 10, line 301: Do the results change by tightening the coincidence criteria?*

Specifically for the SAGE-II comparison the number of collocations is not that large, therefore we prefer not to tighten the criteria here. However, we have checked that even with a reduced number of collocations with SAGE-II we get essentially the same results.

12. *Page 11, line 304: By "mean error", do you mean the standard error of the mean?*

With "mean error" we refer to the mean of the error given in the product. We will clarify this in the text.

13. *In Fig. 10, there seems to be a bias in the background case, the agreement is good mostly between 20 and 25 km with significantly larger biases above and below this altitude range.*

Yes, this is true, We will update the text accordingly.

14. *Page 11 and line 320: Why do the size distribution issues affect only low altitudes? Please discuss.*

The impact of the volcanoes changes the size distribution of the particles. This is mainly limited to the lower altitudes because perturbations in the particle amount and their sizes due to volcanic eruptions rapidly decrease with the altitude. Perturbations due to volcanic eruptions usually do not reach above 20 km in the period from 2002 to 2012.

We will explain this in the text.

15. *What are the black vertical lines in all the panels in Fig. 11?*

    These (grey) lines mark times of degraded instrument performance (like decontamination period or switch-offs). We will mention this in the caption.

16. *Page 12, line 339: Note that the volcano Nabro occurred at low latitude (13° N) and the aerosol plumes spread later to higher latitudes.*

    We will update the text accordingly.

17. *Page 13, lines 374-376: I think the interpretation of the anomalies at altitudes above 25 km in terms of QBO is an interesting result that needs to be discussed further, rather than simply assuming it to be the case. Please add a plot of a QBO index on top of the panels in Fig. 12 so the correlation between the aerosol anomaly and the QBO can be seen more clearly and then discuss the observed anomalies at middle/high latitudes for the easterly and westerly phases of QBO and in terms of aerosol transport from the tropics. Also please discuss the effect in terms of altitude.*

    A detailed discussion on transport effects and QBO is included in our water vapour / methane paper (Noël et al., 2018). This also includes a plot of the QBO index. We do not want to repeat this full discussion in the present paper, but make a reference to this paper and a related one focusing on SCIAMACHY limb data (Brinkhoff et al., 2015, see below) with a short summary and add the QBO index in Fig. 14 (together with the time series at 25 km) for clarification.

[Figure]

Brinkhoff et al. (2015):
Ten-year SCIAMACHY stratospheric aerosol data record: Signature of the secondary meridional circulation associated with the quasi-biennial oscillation, in: Towards an Interdisciplinary Approach in Earth System Science (p. 49–58), Springer, Switzerland, https://doi.org/10.1007/978-3-319-13865-7_6

18. *Do the linear trends shown in Fig. 15 conform to trends from other studies, if any?*

    We do not know of other extinction studies covering the same time, altitude and latitude range. In any case, a comparison of changes would not be straightforward because of the specific spatial/temporal sampling of the SCIAMACHY data.

19. *Page 14, line 413-414: Is the QBO effect expected to be similar for gas species and aerosols?*

    Yes, we see similar effects in our greenhouse gas products, see above.

---

## Author Comment (AC2) · 31 Jul 2020

**Reply to referee 2**

We thank the referee for the overall positive judgement and will consider the comments in the revised version of the paper. In the following, the original reviewer comments are given in *italics*, our answer in normal font.

[Figure]

Answers to comments:

1. *One of the bigger issues, which is already brought up the other reviewer, is the low frequency vertical oscillation of the profiles. This could limit the usability of the data set, and the authors make a rather sweeping assumption that this is due to the nature of the onion peeling method without regularisation. I strongly recommend further work to track down and potentially improve the oscillatory behaviour; if it is as simple as adding regularization to the retrieval, then it certainly should be done.*

   A regularisation is not possible for the current onion peeling method as it would require the coupling / simultaneous retrieval of different tangent altitudes (including lower ones).

   For further information, we repeat here our answer to referee 1:

   We agree with both referees that the vertical oscillations are the most critical issue for the SCIAMACHY solar occultation data product. This is why we explicitly mention it e.g. in the conclusions. These oscillations are not only a problem for the extinction retrieval but also for the greenhouse gas profile retrievals published in earlier studies. We have investigated this issue for several years, but could not identify the reasons for these oscillations. We assume they are caused by a deficiency in the radiometric calibration in combination with the onion peeling method as they seem to appear at all wavelengths. The only way to handle these in the current algorithm is to apply an additional vertical smoothing of the profiles, which we do for trace gas profiles using a boxcar of 4.3 km width. The value of 4.3 km is chosen, because this corresponds to the approximate vertical range of one readout (combination of size of instantaneous field of view and scan). We could choose a larger smoothing width here and/or apply additional smoothing to the extinction / transmission profiles. Since the oscillations have a period of about 10 km, we would need a smoothing width of at least this size, which would

result in a data product with a very low vertical resolution (only ∼2 independent data points). We decided not to do this, as this can still be done by data users if required for a specific purpose.

Note that these oscillations become less prominent (amplitudes <10%) when comparing with data sets where more collocations covering longer times are available (e.g. SCIAMACHY limb and also the newly included OSIRIS data). This averaging effect indicates that sampling and statistics also play a role here.

Our solution to overcome the problem of vertical oscillations is to use anomalies for scientific studies (as we do in the paper). In these anomalies systematic effects – including the oscillations – are essentially removed while keeping the vertical resolution.

We will explicitly include this in the abstract and conclusions of the paper.

2. *The other issue is the reported linear changes that are derived from the time series. The nature of the time series is highly variable due to the volcanic perturbations as nicely shown in Fig 14. The linear analysis is simply not justified. Yes, you can fit a straight line to this, but to do so is not justified, and then to report a "significant positive change of 20–30% per year" is somewhat misleading. Here the comparison with the SCIAMACHY limb scattering retrievals is quite interesting and reasonable, but with differences that the authors claim are due to different measurement times and locations. It would be better to put effort into understanding these differences and skip the linear analysis.*

We agree that due to the volcanic eruptions the temporal evolution of extinction is indeed not linear. We only use a linear model to somehow quantify this change (we explicitly do not call it a trend). We also explicitly mention, e.g. in abstract and conclusions, that the changes we derive at lower altitudes are due to the volcanic eruptions. The numbers we give are therefore very specific for our data set. Nevertheless, we think they are a useful result which should be mentioned.

[Figure]

In order to avoid misunderstanding, we will clarify that these numbers should not be interpreted as a continuous trend.

Regarding the differences to limb, we will add some further explanation, i.e. that limb observations are affected by changes in particle size that might have an influence on the resulting linear changes.

3. *Other smaller issues: Every time that an agreement between observations is claimed to be "good", please be sure to quantify.*

   We will check this and add information where required.

4. *Several times in the paper, the authors refer to the "extinction", where usually it is referring to the aerosol extinction. Please include this each time as extinction is a generalized quantity in radiative transfer and does not just refer to aerosol.*

   Agreed. We will update the text (and the title) accordingly.

5. *The statement at the bottom of page 2 that occultation measures extinction "whereas" limb sounders are more sensitive to smaller particles need qualification. Please explain in more detail. Do you mean limb scattering? In general limb scattering is definitely sensitive to large particles.*

   The sensitivity of limb measurements to smaller particles refers to the particle size distribution, which is not dealt with in this paper. We will delete this part of the sentence to avoid misunderstanding.

6. *For SAGE II comparisons, why include the time criteria of 9 h if it doesn't matter? Also, "temporal distance" is not a standard phrase; "time difference" is clearer.*

   Agreed, 9 h is indeed no limiting factor. We will reformulate this and also replace "temporal distance" by "time difference".

7. *The explanation of figure 3 needs to be clarified on page 5. What is the numerical sun shape function, S? For a localized aerosol or cloud layer, the transmission will*

*have a perturbed shape. How would this be handled by the algorithm to derive the shape function?*

The sun shape function $S$ refers to the sun shape without atmospheric influences. It is determined from the reference measurements at higher altitudes (around 90 km) where influences of aerosol and clouds can be neglected. We will clarify this in the text.

8. *What is the impact of choosing second order for the polynomial in the fit to the transmission spectra?*

Since the fitting windows are usually not that large it is sufficient to use a small order polynomial to describe the background signal. However, as long as there are no spectrally broadband absorption features also higher orders do not change the results very much. We tried several orders, and second order seemed to be most appropriate in our case.

9. *Table 1 lists SCIAMACHY and OMPS nadir modes. These are not used for stratospheric aerosol to my knowledge.*

In the table we listed all measurement modes (even if not used for stratospheric aerosol retrieval). We will update Table 1 and mark those modes which are not used for stratospheric aerosol retrieval for clarification.

10. *Figure 3 caption uses the word "spectra" for the figure. These are not spectra.*

Yes, this wording is wrong. We will correct this.

11. *Figure 6 caption should explain the terms in the fit.*

Will be done.
* * *

---

## Author Response (AR2)

**Stratospheric Extinction Profiles from SCIAMACHY Solar Occultation**
by S. Noël et al.

MS No.: acp-2020-113

**Authors' Response**

**Reply to comment of Editor/Reviewer 2 to Revised Version**

**Comment:**

The authors have done a reasonable job of addressing the majority of the reviewer concerns. However, I still find the "linear change" analysis at the end of the paper to be unacceptable. A linear fit to this highly variable time series is not appropriate and should not be used.

**Answer:**

We have removed the "linear changes" analysis from the paper.

**List of changes**

Text related to "Linear changes" and Fig. 17 have been removed.

[revised manuscript text omitted]

~~Largest differences between occultation and limb occur at the lowest altitudes. This is because limb observations are affected by changes in particle size caused by volcanic eruptions that might have an influence on the resulting linear changes. At 15 a significant positive change of 20–30% per year is derived, which decreases with increasing altitude and becomes insignificant above about 22–25 . Higher altitudes are mainly affected by QBO related variations which mostly average out. The change at lower altitudes is dominated by the increased aerosol load due to the volcanic eruptions during the second half of the time series (see Fig. 16). These changes should therefore not be interpreted as a continuous trend. In general, the derived values are very specific for the analysed data sets and their temporal and spatial sampling.~~

[revised manuscript text omitted]